ecology, health and disease and epidemiology, immunology

nutrition, gastrointestinal helminths, *Heligmosomoides polygyrus*, host–parasite interactions, anthelmintic treatment, *Apodemus sylvaticus*

**Author for correspondence:**
Amy R. Sweeny
e-mail: a.r.sweeny@ed.ac.uk

†These authors contributed equally to this study.

# Supplemented nutrition decreases helminth burden and increases drug efficacy in a natural host–helminth system

Amy R. Sweeny[1], Melanie Clerc[2], Paulina A. Pontifes[3],
Saudamini Venkatesan[1], Simon A. Babayan[4,†] and Amy B. Pedersen[1,†]

[1]Institute of Evolutionary Biology and Centre for Immunity, Infection and Evolution, School of Biological Sciences, University of Edinburgh, Edinburgh EH9 3FL, UK
[2]MRC Centre for Inflammation Research, The Queen's Medical Research Institute, University of Edinburgh, Edinburgh EH16 4TJ, UK
[3]Departamento de Etología, Fauna Silvestre y Animales de Laboratorio, Facultad de Medicina Veterinaria y Zootecnia, Universidad Nacional Autónoma de México, Avenida Ciudad Universitaria 3000, CP 04510 Coyoacán, Ciudad de México, México
[4]Institute of Biodiversity Animal Health and Comparative Medicine, University of Glasgow, Glasgow G12 8QQ, UK

ARS, 0000-0003-4230-171X; MC, 0000-0003-1195-4558; PAP, 0000-0002-7334-8070; SV, 0000-0003-2322-3170; SAB, 0000-0002-4949-1117; ABP, 0000-0002-1385-1360

Gastrointestinal (GI) helminths are common parasites of humans, wildlife, and livestock, causing chronic infections. In humans and wildlife, poor nutrition or limited resources can compromise an individual's immune response, predisposing them to higher helminth burdens. This relationship has been tested in laboratory models by investigating infection outcomes following reductions of specific nutrients. However, much less is known about how diet supplementation can impact susceptibility to infection, acquisition of immunity, and drug efficacy in natural host–helminth systems. We experimentally supplemented the diet of wood mice (*Apodemus sylvaticus*) with high-quality nutrition and measured resistance to the common GI nematode *Heligmosomoides polygyrus*. To test whether diet can enhance immunity to reinfection, we also administered anthelmintic treatment in both natural and captive populations. Supplemented wood mice were more resistant to *H. polygyrus* infection, cleared worms more efficiently after treatment, avoided a post-treatment infection rebound, produced stronger general and parasite-specific antibody responses, and maintained better body condition. In addition, when applied in conjunction with anthelmintic treatment, supplemented nutrition significantly reduced *H. polygyrus* transmission potential. These results show the rapid and extensive benefits of a well-balanced diet and have important implications for both disease control and wildlife health under changing environmental conditions.

## 1. Introduction

Gastrointestinal (GI) helminth infections are ubiquitous in nature and among the most common causes of chronic disease in wildlife, livestock, and human populations [1]. Helminth infections are associated with a range of clinical morbidities including stunted development and cognition, and impaired physical condition and productivity [1–5]. Among wildlife, helminth infections can significantly impact host survival and reproduction, and thereby play a key role in regulating population dynamics [6–8]. To reduce the burden (number of worms per individual) of helminth infections, standard treatment in humans and livestock is drug therapy [9,10]. However, despite the high availability and low cost of anthelmintic drugs, morbidity from helminth infections remains high [2], reinfection post-treatment is rapid [11], and drug resistance is spreading [12,13].

High reinfection rates of GI nematodes, the most common helminths [2], are due in part to transmissible stages that can persist for long periods of time in the environment [14]. In human populations where worms are endemic, within 1 year of anthelmintic treatment *Ascaris lumbricoides* prevalence can rebound to nearly 100% of pre-treatment levels [15,16]. Effective helminth control, therefore, requires not only reducing burdens within individuals, but also reducing exposure and susceptibility to reinfection. Understanding the environmental and host factors that drive susceptibility to reinfection is crucial both for informing infection control and for understanding how fluctuating environmental conditions may influence helminth dynamics in natural populations [17].

Resource availability has been implicated as an important underlying factor that can alter responses to infection and treatment [18,19]. Micronutrient, macronutrient, and energy deficiencies can impair the immune system [20] and insufficient resources for costly immune responses can worsen the consequences of nematode infection [21]. This is evident in humans where pre-existing malnutrition in areas of poor nutrition can worsen nematode infection outcome [18,22] and in livestock where the increased resource demands of late pregnancy and lactation are often associated with a substantial increase in GI nematode burdens [23]. Nutritional supplementation is predicted to alleviate trade-offs between energetically costly processes (e.g. reproduction, body condition, or immunity) that can arise under conditions of limited resources, and, therefore, may reduce susceptibility to reinfection after treatment [24,25]. However, in practice, results remain equivocal; a recent meta-analysis reviewing clinical trials of single or combined micronutrient supplements showed mixed effects of nutrition supplementation on nematode infections, highlighting the lack of clarity in natural populations [26].

Despite a vast body of knowledge investigating mechanistic links between nutrients and nematodes in the laboratory, translation to natural populations is challenging. Laboratory mouse models have provided key evidence that both macro- and micronutrients can play a key role in immunity to nematodes and susceptibility to infection [27–31]. For example, protein [29] and zinc [27,28] deficiencies have been shown to increase worm burdens while reducing eosinophilia and parasite-specific IgG1 response [27] to *Heligmosomoides bakeri* (formerly *Heligmosomoides polygyrus bakeri* [32]), a well-studied model nematode. However, laboratory conditions are highly controlled with standard, invariant diets, age-matched and often single-sex cohorts of inbred lines and, therefore, are unlikely to mimic life in the wild. Furthermore, there is an increasing appreciation for the complexity of outcomes of resource supplementation in the wild. Increased resource availability can improve host condition and immunity resulting in reduced infection, but may also alter host behaviour and aggregation around food sources such that transmission is increased [33]. Although supplementation experiments have also been investigated in wild mouse models [7,34,35], these studies augmented resources of the same type as was available in the environment (e.g. seeds) rather than introducing supplemental food with additional micro- and macronutrients. It, therefore, remains unclear how whole-diet supplementation affects immunity to helminths, drug treatment efficacy, and post-treatment recrudescence of infection in natural populations.

Here, we experimentally enriched nutrition in a wild population of wood mice (*Apodemus sylvaticus*) with a well-balanced diet, to test the impacts on resistance to *H. polygyrus* and anthelmintic treatment efficacy under ecologically realistic conditions. Wood mice live in woodlands across much of Europe and are chronically and commonly infected with *H. polygyrus* (prevalence 20–100%) [36,37], a sister taxa to *H. bakeri* [38]. While anthelmintics can significantly reduce infections in wood mice, reinfection to pre-treatment burdens typically occurs within two to three weeks [39,40]. Further, wood mice, like most wild animals, have substantial and simultaneous energetic demands for reproduction, foraging, and survival [41,42], conditions which laboratory settings cannot replicate, but which likely impact infection exposure, immunity, and resource allocation. Crucially, here we have the unique ability to test the same host–helminth system in both the wild and the controlled conditions of our wild-derived colony of wood mice in order to control infection/reinfection, exposure, co-infection, and other important factors. We used paired experiments in wild and laboratory populations of the same species to test the effects of supplemented nutrition and anthelmintic treatment on (i) *H. polygyrus* burden and egg shedding and (ii) body condition and immune responses. We use data from the laboratory population to better infer mechanisms in a controlled setting, and data from the wild for the translation to an ecologically realistic setting. In both settings, we found strong evidence of rapid and broad impacts of this well-balanced diet for host condition and helminth resistance. Our results suggest that whole-diet supplementation could provide significant benefits for helminth control by increasing the host's ability to respond to infection and reducing the probability of reinfection.

## 2. Materials and methods

### (a) Ethics statement

All animal work was carried out under the approved UK Home Office Project License 70/8543 in accordance with the UK Home Office in compliance with the Animals (Scientific Procedures) Act 1986 and approved by the University of Edinburgh Ethical Review Committee. Fieldwork was carried out with permission of the Forestry Commission Scotland, permit reference SUR09.

### (b) Field experiment

We conducted the field experiment in a woodland wood mouse population naturally infected with *H. polygyrus* [40] located in Falkirk, Scotland (Callendar Wood, 55.990470, −3.766636), during the peak wood mouse breeding season (May–September); when host energetic demands are highest. We used a 2 × 2 factorial design in two temporal replicates of eight weeks (2015/2016), where (i) nutrition was manipulated at the population (trapping grid) level, high-quality whole-diet food pellets (hereafter 'diet') versus control (unmanipulated), and (ii) anthelmintic treatment (hereafter 'treatment') was manipulated at the individual level, control (water) versus treatment (see electronic supplementary material for full details).

Grids were supplemented for two weeks before and then throughout the eight-week experiment twice per week with a homogeneous scattering of 2 kg/1000 m$^2$ of sterilized Trans-Breed™ mouse chow pellets (nutritional information detailed in electronic supplementary material, table S1) to complement natural food availability. Mice were then live-trapped three nights/week using Sherman live traps (H.B. Sherman 2 × 2.5 × 6.5 inch folding trap, Tallahassee, FL, USA). At first capture, mice weighing greater than 10 g were tagged with a subcutaneous microchip

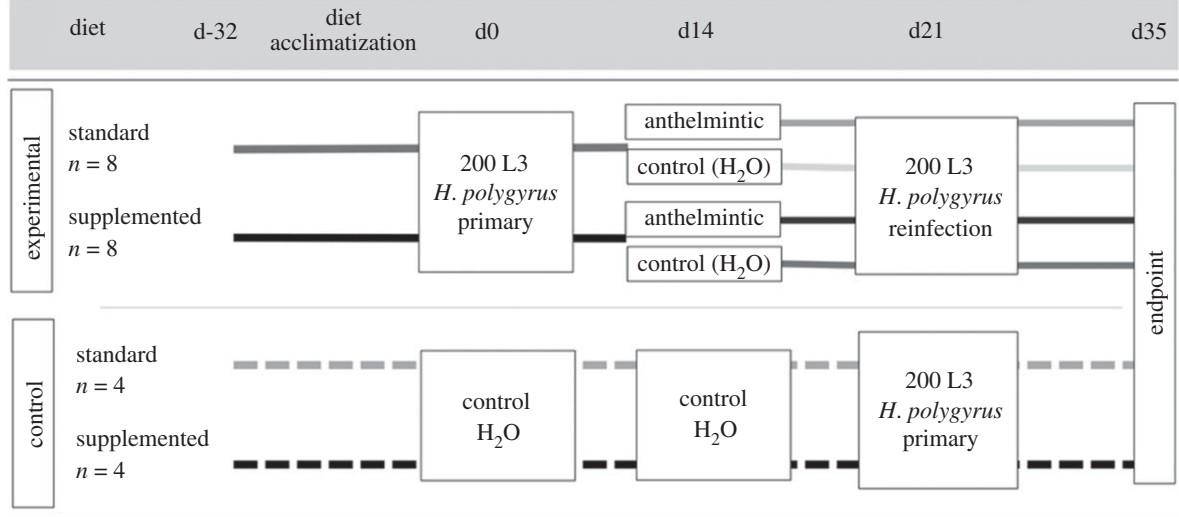

**Figure 1.** Diagram of the laboratory experimental design. Mice were assigned to diet groups 32 days before infection (d-32). After diet acclimatization, all experimental (n = 16; solid lines) mice were given a primary challenge of *H. polygyrus* (d0). On day 14 post-infection, experimental mice were randomly assigned within diet groups to receive an anthelmintic treatment (darker lines) or a control dose of water (lighter lines; n = 4/ group). On day 21, all experimental mice received a secondary challenge of *H. polygyrus*, and were culled on day 35, 14 days post-secondary challenge. On days 0 and 14, control mice (dashed lines, n = 4/ group) received equivalent volumes of water, and on day 21 received a primary challenge with *H. polygyrus*. Control mice were culled on day 35, 14 days post-primary challenge.

transponder for identification (Friend Chip, AVID2028, Norco, CA, USA) and rotationally assigned within each sex to receive either a control dose of water or a weight-adjusted dose of a combination of Ivermectin and Pyrantel pamoate anthelmintic treatment (electronic supplementary material, §1.1.1).

For each mouse at every capture, we measured morphometric data (sex, age, body mass (g) and length, fat scores, and reproductive status as described in electronic supplementary material, §1.1.2), and collected a blood and faecal sample. Mice captured 12–16 days after first capture (the period of efficacy for this drug combination that we have previously observed in wild wood mice [39,40]) were sacrificed for additional destructive sampling. Mice caught beyond this date range, or those pregnant or lactating, were not sacrificed. Eye lenses were collected as a proxy of age (see electronic supplementary material, §1.1.2) and the small intestine, caecum, and colon of each individual were collected for counts of adult *H. polygyrus* worms from all sacrificed animals.

### (c) Laboratory experiment

We conducted a 2 × 2 factorial design in a laboratory colony of *A. sylvaticus* (details in electronic supplementary material, §1.2.1) to parallel the field experiment; both (i) diet and (ii) anthelmintic treatment (control (water) versus treatment) were manipulated at the individual level (figure 1). Transbreed™ was used for the diet supplementation. Rat Mouse 1 (RM1™) was the control diet as it a commonly used maintenance diet which contains lower nutrients, but is not considered a restrictive diet (electronic supplementary material, table S1). All mice were fed ad libitum and were given a 32-day diet acclimatization period. Sixteen mice aged 15–21 weeks (median 18 weeks) were randomly assigned to the four experimental groups (n = 4/group; figure 1): (i) supplemented nutrition, treated, (ii) supplemented nutrition, control, (iii) standard nutrition, treated, and (iv) standard nutrition, control. Eight mice were designated as controls and placed on the same diets as experimental mice (n = 4 per group; figure 1). We performed primary and secondary *H. polygyrus* inoculations to mimic the high level of exposure and reinfection found in wild wood mice (figure 1; electronic supplementary material, §1.2).

Once per week, body weight and fat scores were recorded. Weekly blood samples for each individual were collected via venesection (tail bleed) on days 0, 14, 25 and via venepuncture (cheek bleed) on day 21. Individuals were sacrificed and sampled

on day 35 as described above. Faecal samples were collected three times/week for the duration of the experiment.

### (d) Laboratory assays for both field and laboratory experiments

We measured *H. polygyrus* shedding as eggs per gram (EPG) of faeces (EPG) using salt flotation [39] (details in electronic supplementary material, §1.3.1). Briefly, *H. polygyrus* eggs were counted and standardized by the weight of the sample to estimate EPG. EPG values were rounded to the nearest integer for subsequent analysis. We used ELISA assays to measure (i) total faecal IgA concentration and (ii) sera *H. polygyrus*-specific IgG1 antibody titres for each mouse at each sampling point as previously described [43] (details in electronic supplementary material, §1.3.2). We calculated total faecal IgA concentration by extrapolation from a standard curve of known concentrations from a synthetically manufactured standard antibody. *Heligmosomoides polygyrus*-specific IgG1 was calculated as a relative concentration to a positive reference sample. We refer to both IgA and IgG1 values as 'antibody concentration'.

### (e) Statistical analyses

We carried out all statistical analysis using R v. 3.6.1 [44]. All models were fitted using the package 'glmmTMB', with the exception of fat score models which have ordinal response variables and were fitted as cumulative link mixed models using the package 'ordinal'. Model formulae are listed in electronic supplementary material, table S2. *Post hoc* comparisons of interaction levels were calculated using the package 'emmeans'. All models including multiple samples per individual included mouse ID as a random effect. A grid-by-year interaction as a random effect was tested in all wild models to account for possible variation due to experimental design but in all cases was associated with negligible variance.

### (i) Analysis of *Heligmosomoides polygyrus* infection

To investigate the impact of supplemented nutrition on *H. polygyrus*, we used generalized linear models (GLMs) or generalized linear mixed-effects models (GLMMs) with negative binomial (NB) error families. Wild models were fitted to the following response variables: (i) intensity of infection (EPG) at first capture (before drug treatment), (ii) mean EPG per individual for subsequent post-treatment captures, and (iii) infection burden

(adult worm count) at final capture. Because few mice were captured beyond the 12–16 day range for endpoint, and those that were skewed towards reproductive females who were not sacrificed, EPG data for post-treatment captures were restricted to timepoints up to 16 days post-treatment. Fixed effects in all models included diet and host characteristic variables (electronic supplementary material, table S2), and models (ii) and (iii) included fixed effects of drug treatment and a treatment-by-diet interaction. Age was only included as an explanatory variable in the worm burden model for sacrificed animals, where eye lens weight allowed estimation of age [45]; in EPG models, body mass represents a less-resolved approximation of age. Diet group was classified as 'supplemented' if greater than 50% of captures were on supplemented grids and as 'control' otherwise. However, we fitted another set of three models including three levels of diet as an explanatory variable (control, mix, supplemented), where 'mix' represented mice found on both grid types ($n = 16$) and confirmed that effects of supplemented nutrition were not dependent on time (fraction of captures) spent on grid type (electronic supplementary material, §2).

Laboratory GLMs were fitted to the following response variables: (i) peak EPG shed, (ii) total EPG shed, and (iii) adult worm burden. Although all experimental mice experienced primary and secondary infection, only two individuals had EPG values greater than 0 after reinfection, and so models (i) and (ii) represent primary infection only. Additionally, although anthelmintic drugs were administered to half of the experimental group before secondary challenge, there was no difference in worm clearance (as indicated by EPG) between drug-treated and control mice and they were combined within diet groups for these analyses. Explanatory variables for all models are listed in electronic supplementary material, table S2, and included diet, host characteristics, and day of experiment as fixed effects.

### (ii) Analysis of body condition and immunity

We investigated the effect of diet on two metrics of body condition in the wild and laboratory (i) body mass (g) and (ii) total fat score (sum of dorsal and pelvic fat scores [46]) using GLMMs with Gaussian error distributions (body mass) and cumulative link mixed models (total fat score). Diet and drug treatment, host characteristics, time, and H. polygyrus infection intensity (log of egg/gram + 1) were included as fixed effects (electronic supplementary material, table S2). Reproductive status and body length were included as covariates in wild models to account for variation in body size. We used the same fixed effects for the laboratory models as in the wild with the exception of reproductive status which was not applicable and body length as absolute age was available (electronic supplementary material, table S2).

We fitted GLMMs with Gaussian error distributions with either total non-specific IgA or H. polygyrus-specific IgG1 (for standardized IgG1 values greater than 0 only) to test the effect of diet and H. polygyrus infection on antibody response in the wild and laboratory. Wood mice in the laboratory had little-to-no specific IgG1 antibody levels until approximately 21 days post-infection (dpi) so models were limited to blood samples collected during secondary challenge (28 and 35 dpi). There were not enough samples per individual for model convergence with individual as a random effect so models were fitted to the mean values of IgG1 for each individual. Fixed effects included host characteristics, year and day of experiment, and experimental manipulations (electronic supplementary material, table S2). The residuals of a body weight-by-body length linear regression with Gaussian error were used as a fixed effect representing body condition index.

## 3. Results

We captured 91 individual mice 310 times throughout our field experiment (2015: $n = 49$ and 2016: $n = 42$), 61 of which were captured greater than 1 time (mean capture number = 3.42 ± 0.26).

Of these, 35 mice were sacrificed after two weeks to measure H. polygyrus worm burdens. For all other captures, EPG represents a proxy of worm burden (Pearson's r: $r_{2015} = 0.72$, $r_{2016} = 0.80$; electronic supplementary material, figure S3).

### (a) Supplemented nutrition decreased Heligmosomoides polygyrus worm burdens and egg shedding

On average, mice had access to supplemented diet for approximately 30 days (range 12–63 days). At first capture, mice on supplemented grids had significantly lower H. polygyrus EPG than mice on control grids (figure 2a, $\beta = -2.47$, $p < 0.001$), with approximately 88% less egg shedding than mice that only had access to their normally available sources of nutrition (electronic supplementary material, table S4). The significant benefit of supplemental nutrition continued, as mice on supplemented grids caught 12–16 days after their first capture also had 60% fewer adult worms compared to mice on control grids (figure 2c, $\beta = -1.20$, $p = 0.045$; table 1). Furthermore, models specifying an additional diet group for individuals who were captured on both supplemented control grids showed that benefits of supplementation were conferred even for transient exposure to higher quality food (electronic supplementary material, figures S1 and S4). We also found that larger mice had higher infection intensity ($\beta = 0.19$, $p = 0.045$) at first capture, and that larger ($\beta = 0.21$, $p = 0.019$) and older ($\beta = 2.22$, $p = 0.016$) individuals had higher worm burdens (electronic supplementary material, table S5) at the experimental endpoint. In addition, reproductively active individuals had lower infection intensities at first capture ($\beta = -1.63$, $p = 0.012$).

In the laboratory experiment, supplemented diet reduced both peak ($\beta = -1.09$, $p = 0.017$) and total H. polygyrus EPG ($\beta = -1.07$, $p = 0.030$) compared to mice on the standard diet, and reduced shedding (EPG) to zero following reinfection (figure 2d,e; electronic supplementary material, table S4). While there was no difference in adult worm burdens between mice on supplemented or control diets after primary infection, mice on the supplemented diet were significantly less susceptible to secondary challenge, with 75% lower adult worm burdens ($\beta = -1.76$, $p = 0.002$, figure 2f; electronic supplementary material, table S4).

### (b) Supplemented nutrition improved anthelmintic drug efficacy

We found a significant interaction between diet and anthelmintic treatment on intensity of infection (EPG; $\beta = -4.51$, $p = 0.014$, table 1). In the wild, treatment reduced shedding to less than 1 H. polygyrus EPG faeces in diet-supplemented mice for two weeks following treatment, while treated mice on control grids still shed approximately 29 EPG during the same period (Tukey post hoc test: $\beta = -6.06$, $p < 0.001$, figure 2b). Likewise, although anthelmintic treatment significantly reduced worm burden for all mice ($\beta = -2.74$, $p < 0.001$), efficacy was highest in mice on supplemented grids (Tukey post hoc test: $\beta = -3.46$, $p = 0.017$, figure 2c), resulting in complete worm clearance for all but one mouse that had a single worm (figure 2c; electronic supplementary material, table S4).

### (c) Supplemented nutrition improved wood mouse condition and immunity

Wild wood mice on supplemented grids had higher body mass and total fat scores (FS) compared to mice on control

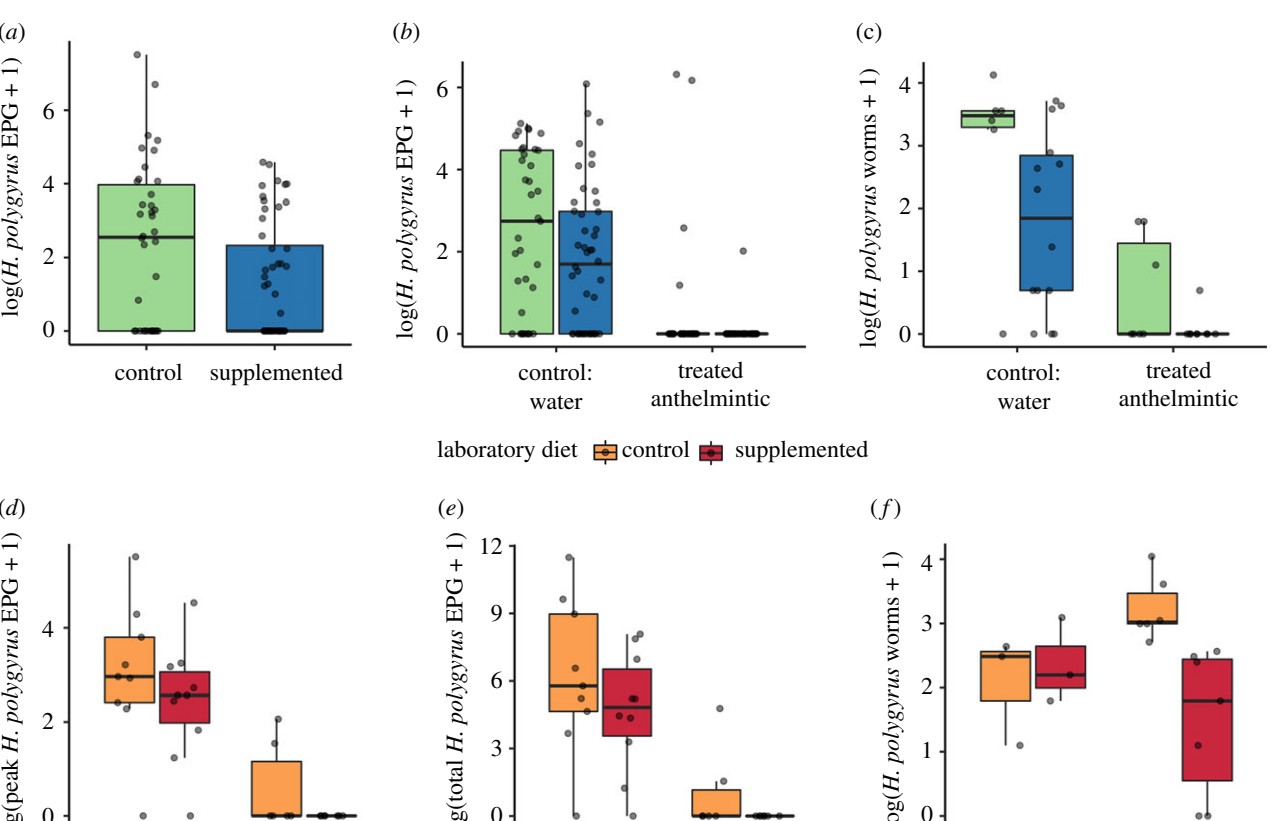

**Figure 2.** Effect of supplemented nutrition on *H. polygyrus* infection in wild and laboratory wood mice. Mice on supplemented grids compared to control grids had lower mean *H. polygyrus* abundance (log EPG + 1) (*a*) at first capture, *N* = 91 individuals and (*b*) after treatment (*N* = 62 individuals; 166 captures) and had lower worm burdens (log of worm count) (*c*) at endpoint for culled individuals (*N* = 36). Mice in the laboratory on a supplemented versus standard (control) diet had lower (*d*) peak and (*e*) total *H. polygyrus* EPG during primary and secondary challenge and had no difference in (*f*) worm burden (log of worm count) 14 days primary infection, but reduced worm burdens 14 days after secondary challenge. *N* = 19 mice (Primary only control group, *N* = 6, Primary; Secondary group, *N* = 13). Grey points represent individual mice, jittered to reduce overlap between points. (Online version in colour.)

grids (mass $\beta$ = 1.69, $p$ = 0.006; FS $\beta$ = 01.12, $p$ = 0.003; figure 3). In addition, wild mice from 2016 had higher body mass ($\beta$ = 1.13, $p$ = 0.042) but lower FS ($\beta$ = −1.14, $p$ < 0.001) than those in 2015 (electronic supplementary material, table S7). In the laboratory, mass did not significantly vary between diet groups (table S7), but supplemented mice had higher fat scores compared to control mice (FS $\beta$ = 5.02, $p$ = 0.047). Laboratory males had both higher mass ($\beta$ = 8.53, $p$ < 0.001) and FS ($\beta$ = 15.07, $p$ < 0.001) than females (electronic supplementary material, table S7).

In the wild mice, anthelmintic treatment ($\beta$ = 1.91, $p$ = 0.039; electronic supplementary material, table S8) and body condition were positively associated with higher concentrations of IgA in both years ($\beta$ = 0.43, $p$ = 0.001, figure 4*b*). Body condition was the only significant predictor of *H. polygyrus*-specific IgG1, where better body condition was associated with higher antibody concentrations ($\beta$ = 0.02, $p$ = 0.022, figure 4*c*; electronic supplementary material, table S8). Total faecal IgA antibody levels were lower in 2016 ($\beta$ = −5.14, $p$ < 0.001). Wood mice on supplemented grids had significantly higher total faecal IgA antibody concentrations in 2016 ($\beta$ = 6.31 $p$ < 0.001, no difference in 2015; figure 4*a*; electronic supplementary material, table S8). In the laboratory, mice on a supplemented diet had both significantly higher total faecal IgA 2–4 weeks post-infection ($\beta$ = 2.40, $p$ = 0.018) and *H. polygyrus*-specific IgG1 after

three weeks post-infection ($\beta$ = 0.20, $p$ < 0.001, figure 4*d*,*e*; electronic supplementary material, table S8).

## 4. Discussion

In this study, we found that supplemented nutrition had dramatic and fast-acting benefits for host resistance, anthelmintic treatment efficacy, body condition, and immunity. By conducting parallel experiments with the same host and helminth species in the wild and controlled laboratory conditions, we were able to test real-world interactions between diet, anthelmintic treatment efficacy, and immunity, and to overcome many of the limitations of field experiments such as variation among individuals in parasite exposure, demographic characteristics, and nutritional status. These results highlight broad benefits of supplemental nutrition in a wild host–helminth system with implications for both helminth and wildlife health management programmes.

Our findings support previous evidence in this system that a combination anthelmintic is highly efficacious [40] and further show that nutritional supplementation can have synergistic impacts on the control of helminths in natural populations. Effective helminth control in endemic areas is difficult because even with readily available anthelmintic drugs, reinfection rates are usually high [11]. Limiting infection in a population requires

**Table 1.** Effects of supplemented nutrition on helminth infection, body condition, and immunity. Italic in table 1 indicates significance at the $p < 0.05$ threshold.

wild

| measure | $N_{obs}$ | main effect: supplement | | | interaction: supplement × anthelmintic | | |
|---|---|---|---|---|---|---|---|
| | | coefficient | s.e. | p-value | coefficient | s.e. | p-value |
| *H. polygyrus*, first capture (EPG) | 91 | −2.47 | 0.60 | *<0.001* | | | |
| *H. polygyrus*, trapping duration (mean EPG) | 62 | −1.56 | 1.07 | 0.145 | −4.51 | 1.84 | *0.014* |
| *H. polygyrus*, endpoint (N worms) | 36 | −1.20 | 0.60 | *0.045* | −2.25 | 1.53 | 0.142 |
| condition, body weight (g) | 202 | 1.69 | 0.62 | *0.006* | | | |
| condition, total fat score | 202 | 0.65 | 0.22 | *0.003* | | | |
| immunity, total IgA | 207 | −0.31 | 1.21 | 0.800 | | | |
| immunity, *H. polygyrus*-specific IgG1 | 121 | −0.10 | 0.09 | 0.282 | | | |

laboratory

| measure | $N_{obs}$ | main effect: supplement | | | interaction: supplement × infection challenge | | |
|---|---|---|---|---|---|---|---|
| | | coefficient | s.e. | p-value | coefficient | s.e. | p-value |
| *H. polygyrus*, primary challenge (Peak EPG) | 19 | −1.09 | 0.46 | *0.017* | | | |
| *H. polygyrus*, primary challenge (Sum EPG) | 19 | −1.07 | 0.49 | *0.03* | | | |
| *H. polygyrus*, endpoint (N worms) | 19 | 0.21 | 0.48 | 0.663 | −1.76 | 0.57 | *0.002* |
| condition, body weight (g) | 90 | 1.63 | 1.47 | 0.268 | | | |
| condition, total fat score | 66 | 0.85 | 0.36 | *0.019* | | | |
| immunity, total IgA | 36 | 2.4 | 1.01 | *0.078* | | | |
| immunity, *H. polygyrus*-specific IgG1 | 13 | 0.2 | 0.05 | *<0.001* | | | |

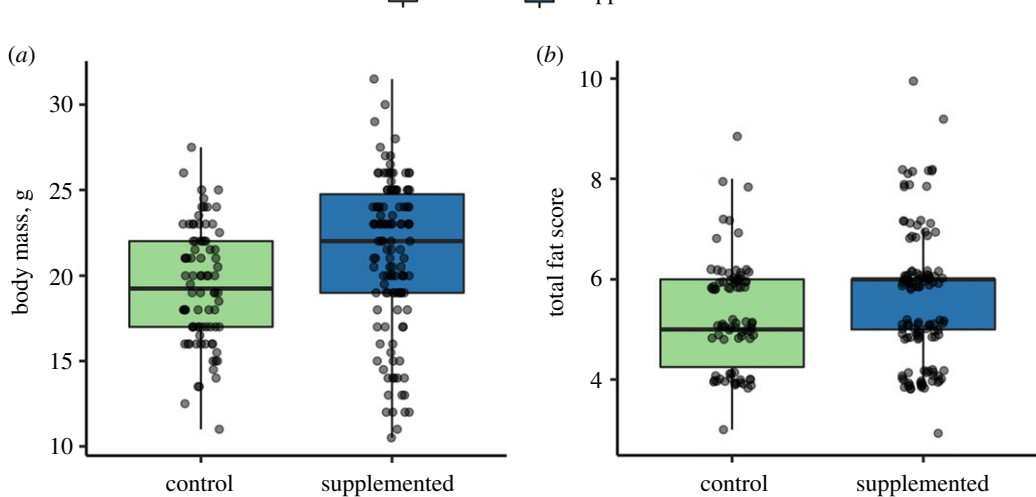

**Figure 3.** Effect of supplemented diet on body condition in the field experiment. Body condition was higher in supplemented individuals ($N = 91$ individuals, $N = 202$ captures) as measured by both (*a*) body mass (g) and (*b*) total fat scores. Points represent raw data jittered to reduce overlap between points. (Online version in colour.)

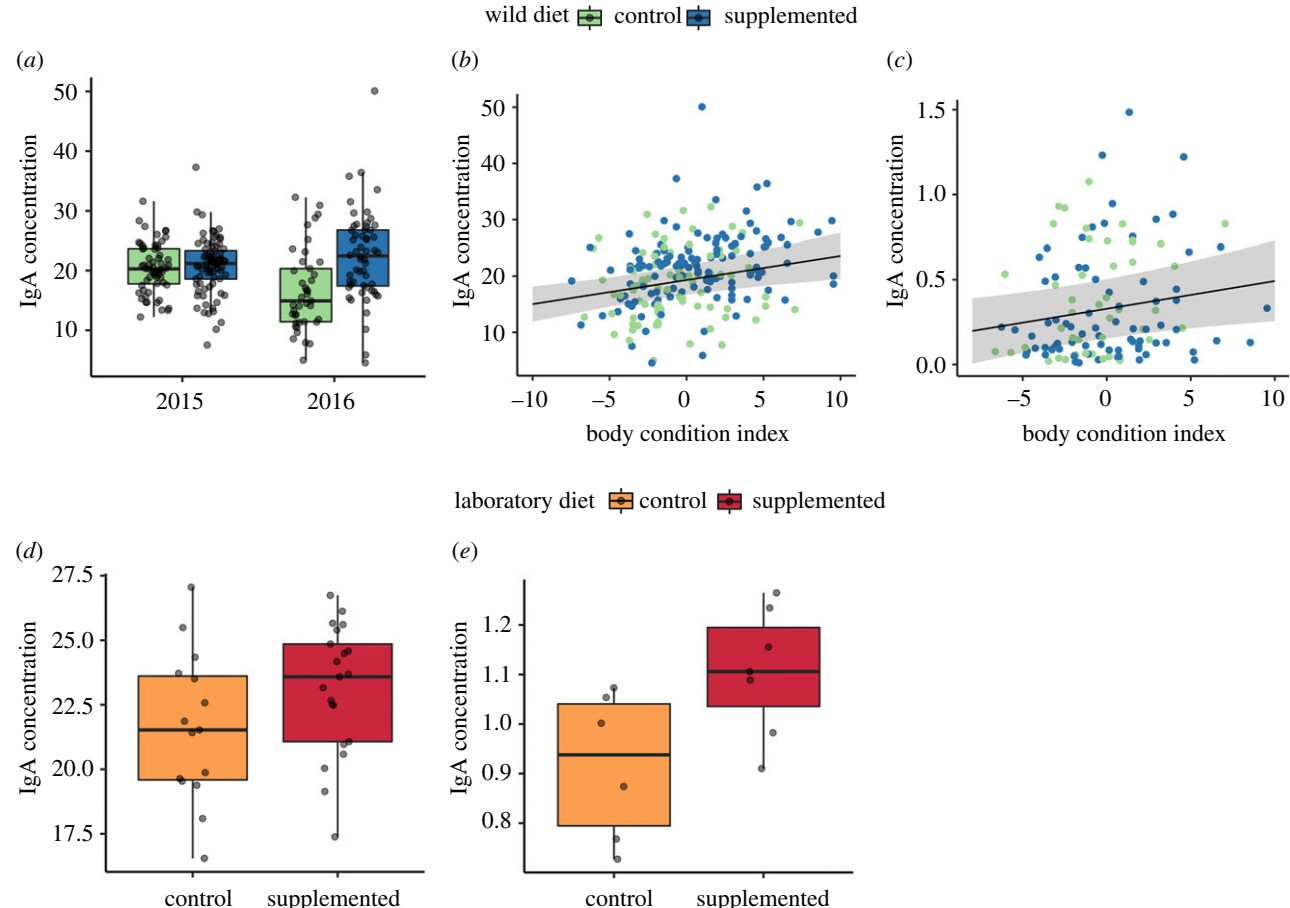

**Figure 4.** The impact of diet on wood mouse antibody responses, wild: top row; laboratory: bottom row. IgA concentration (absolute, ng μl$^{-1}$) at all captures (*a*) across years and (*b*) compared to the body condition index (BCI, weight versus length residuals). (*c*) *Heligmosomoides polygyrus*-specific IgG1 concentration (standardized) at all captures compared to BCI. (*d*) Mean IgA concentration days 14, 21, and 28 post-primary challenge. (*e*) Mean IgG1 concentration during secondary challenge (days 28 and 35). Points represent raw data points, jittered to reduce overlap between points. Point-line plots represent raw data points and model-predicted regression slopes with 95% credibility interval ribbons. (Online version in colour.)

(i) lowering worm burdens, (ii) reducing onward transmission, and (iii) preventing reinfection. In our wild population, we found that supplementation achieved all three: anthelmintic treatment in conjunction with higher quality diet reduced *H. polygyrus* adult worm burdens and egg shedding to almost

zero on supplemented grids. This was replicated in the laboratory, where supplemented individuals shed no eggs during secondary challenge despite harbouring adult worms. Finally, the increased resistance to reinfection in supplemented individuals in the laboratory highlights an additional benefit of

nutrition in managing nematodes in natural populations where post-treatment reinfection is common. This result in an ecologically realistic experimental context provides complementary evidence to previous work showing conflicting results of nutrition supplementation on reinfection rates [26].

The rapid increase in resistance to helminth infection following diet supplementation in paired wild and laboratory experiments provides field-based evidence for a relationship which has previously been reported primarily in model systems. Although higher quality nutrition is often expected to increase immune system performance, confounding of condition and behavioural responses in the wild [47,48] as well as interindividual variation in helminth responses to host resource fluctuation [49] render predicting the net outcome of supplementation difficult. Evidence for variation in the outcome of resource provisioning to date is largely from observational data following access to novel diets due to anthropogenic influence [33] or experimental manipulations that mimic naturally available food fluctuations. For example, supplementation of wood mice (A. sylvaticus) during winter with grass seeds led to a reduction in pinworms (Syphacia stroma and S. frederici), but not other helminth species [34]. Similarly, supplementation of Permoyscus spp. with seeds in conjunction with helminth removal improved host condition and survival [7]. These studies highlight the far-reaching consequences of natural resource fluctuations, but are difficult to compare to studies in captivity where phenotypic responses may vary [50] and to targeted nutrient manipulation for health or treatment purposes.

By confirming results observed in the wild in a laboratory experiment where exposure and individual differences were controlled, we demonstrate that the benefits of whole-diet supplementation are not due to differences in wild wood mice behaviour or foraging patterns, but likely represent an increased ability to respond to infection. We found that supplementing with high-quality diet improved body condition in wild wood mice, but had only a modest effect in our laboratory experiment. This is contrary to previous experimental studies in laboratory mice which found that animals given lower protein had significantly lower weights [27]. This is likely due to two factors: first, animals in the wild have higher energy requirements for immunity, reproduction, thermoregulation, and other processes, and, therefore, may be in worse condition [21]. Second, the control diet in this study was not restrictive in calories nor did it lack particular macro- or micronutrients, instead it served as a baseline maintenance diet, against which we could test an enriched overall diet. By contrast, nutrition manipulation in the laboratory often contains dramatic differences in nutritional content: i.e. protein restriction (2–3% protein) compared to high protein (16–24% protein) [27,30,31,51,52]. The complex and multidimensional nature of the relationship between nutrient concentration and immunity [53] makes direct comparison to previous work manipulating single nutrients difficult. However, in light of the modest differences in macro- and micronutrients in laboratory diet groups, we find a high magnitude of effect without requiring dramatic increases in any one specific nutrient. This finding is further strengthened by our results in the wild, where, as early as 14 days after supplementation and for hosts on supplemented grids for only a portion of captures, an enriched diet appeared to impact the host's ability to mount a more effective protective response to H. polygyrus.

Hosts need adequate levels of macro- and micronutrients for functioning cellular and humoural immune systems [20,54], and antibodies play an important role in immune response to helminths [55]. Faecal IgA is an important component of resistance to GI nematodes and has been used as an indicator of general gut health [43,56]. Parasite-specific IgG1 has a key role in the strong Th2 immune response induced by H. polygyrus [57] through involvement with blocking the maturation of larvae into adult worms within the host intestinal tissue and reducing worm fecundity [58]. Our results align with previous work suggesting that inadequate levels of nutrients (e.g. protein and zinc) can compromise both general and specific host immune responses in mice [27–29,59]. Although our field experiment shows only weak evidence for a direct effect of supplemented nutrition on antibody production, we found positive associations between the body condition index and both total faecal IgA and the H. polygyrus-specific immune response. Therefore, we suggest that the improved body condition of supplemented individuals may result in an indirect effect of supplementation on antibody levels and increased helminth resistance. Although mice with higher mass had higher worm burdens at final capture in the wild, results from the laboratory population suggest that nutrition effects on host response greatly reduces EPG even when worms are present. Typically, immune measures in the wild are difficult to interpret due to the context-dependency of immune phenotypes, and the limited ability to relate immune measures to exposure/infection [42]. Thus, future work will benefit from additional longitudinal data on antibody responses to diet changes in the wild. However, our exposure-controlled laboratory study conducted here on the same host–helminth systems shows evidence of higher specific and general antibody concentrations in supplemented wood mice, and suggests that nutritional availability may be an important factor that drives immune responses and helminth resistance in the wild.

Our study presents clear experimental results from a unique pairing of wild and controlled laboratory studies of the same host–helminth system that nutrition has a rapid, dramatic impact on helminth resistance and should be considered as a viable option for complementing helminth control interventions and conservation efforts. Given the equivocal results from wildlife data detailing responses to provisioning [33] and human clinical trials exploring nutrition supplementation in helminth management [26] and the range of macro- and micronutrients implicated in impacting immunity to GI nematodes [52,60,61], additional work to characterize the mechanisms of the effects of an enriched diet on host immunity as well as the long-term effects of nutrition supplementation in natural populations will be important for understanding their consequences for host health and disease transmission.

Ethics. All animal work was conducted in accordance with the UK Home Office in compliance with the Animals (Scientific Procedures) Act 1986, was approved by the University of Edinburgh Ethical Review Committee, and was carried out under the approved UK Home Office Project License 70/8543. Animal sacrifice was performed using Schedule 1 Methods (Field: Cervical dislocation; Laboratory: Rising $CO_2$ exposure). Fieldwork was carried out with permission of the Forestry Commission Scotland. Permit reference SUR09.

Data accessibility. Data for this manuscript are available from the Dryad Digital Repository: https://doi.org/10.5061/dryad.sbcc2fr5c [62].

**Authors' contributions.** A.R.S., P.A.P., S.A.B., and A.B.P. designed the field studies. A.R.S., A.B.P., and S.A.B. conceived the laboratory study. All authors were involved in the collection of field data. A.R.S., M.C., P.A.P., and S.V. performed the laboratory analyses. A.R.S. performed the statistical analyses and wrote the manuscript. All co-authors contributed comments to the manuscript, approved the final version and agree to be accountable for the contents.

**Competing interests.** We declare we have no competing interests.

**Funding.** This work was supported by PhD studentships from the Darwin Trust of Edinburgh to A.R.S. and S.V., a Torrance Bequest scholarship from the University of Edinburgh awarded to M.C., a Wellcome Trust Institutional Strategic Support Fund (ISSF) grants to A.B.P. (ISSF 2014; J22737) and S.A.B. (097821/Z/11/Z), a targeted

Institute of Biodiversity, Animal Health & Comparative Medicine Research Fellowship to S.A.B., a Wellcome Trust Strategic Grant for the Centre for Immunity Infection and Evolution (095831) to A.B.P. and a University of Edinburgh Chancellors Fellowship to A.B.P.

**Acknowledgements.** We thank the Forestry Commission for permissions for fieldwork in Callendar Park. *Heligmosomoides polygyrus* excretory-secretory product (HES) used in immunological assays was supplied by Rick Maizels (University of Glasgow) and Amy Buck (University of Edinburgh). Thanks also to Greg Albery for insightful comments on the manuscript, and to the many tireless field assistants and volunteers involved in field and laboratory data collection. Lastly, we thank the EGLIDE (Edinburgh, Glasgow and Liverpool Infectious Disease Ecology) group for their feedback and help with the analysis and interpretation of our results.

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
