## [Reviewer comments · Proceedings of the Royal Society B: Biological Sciences]

Review History

RSPB-2020-0859.R0 (Original submission)

Review form: Reviewer 1

Recommendation

Accept with minor revision (please list in comments)

Scientific importance: Is the manuscript an original and important contribution to its field?

Excellent

General interest: Is the paper of sufficient general interest?

Excellent

Quality of the paper: Is the overall quality of the paper suitable?

Excellent

Is the length of the paper justified?

Yes

Should the paper be seen by a specialist statistical reviewer?

No

Do you have any concerns about statistical analyses in this paper? If so, please specify them explicitly in your report.

No

It is a condition of publication that authors make their supporting data, code and materials available - either as supplementary material or hosted in an external repository. Please rate, if applicable, the supporting data on the following criteria.

Is it accessible?

No

Is it clear?

No

Is it adequate?

No

Do you have any ethical concerns with this paper?

No

Comments to the Author

The authors here robustly combine a field and laboratory experiment to factorially test the effects of nutrient supplementation and parasite removal on host immunity and helminth outcomes. The authors demonstrate that supplemental nutrition rapidly improves host resistance to helminths, increases the efficacy of antihelminth treatment, and improves condition and immune defense. This is a very strong study, and I had relatively minor comments about the actual design and analysis. The authors are very transparent about their results, and it was nice to see all the model outputs provided in the supplement.

My only major comment is a more conceptual one. Currently, the manuscript is framed (L81-97) around helminth infections, malnutrition, and supplementation in humans; that laboratory mouse models can inform how nutrients affect infection outcomes; but that wild experiments are needed to better represent the co-evolutionary dynamics of hosts and parasites and natural conditions. Wild mice undoubtedly provide insight into how nutrition impacts parasite outcomes in natural populations, but it isn't entirely clear how that would then translate back to human interventions per se. In general, I think the authors could also add the justifying point (in both the Introduction and Discussion) that nutritional supplementation is also an intervention worth considering in wildlife themselves (e.g., for species of conservation concern). The results of the current study (rapid improvements to host resistance) also carry important implications for how supplemental food resources could affect infection in wild species that are reservoir hosts of zoonotic pathogens. Some discussion of these contexts would only broaden the relevance of the study for not only humans but also wildlife themselves.

Minor comments:

L126: Can you provide the months of the wood mouse breeding season?

L192: Replace with "Generalized Linear Mixed Models"

L194: Is the negative binomial appropriate for mean EPG? If this is a mean across captures, is it still count data (i.e., integers) or now continuous (i.e., with decimals)? If the resulting means fall into the latter, were numbers rounded to the nearest integer for the NB?

L213: Were mass, fat, IgA, and IgG log-transformed prior to the GLMMs? That would prevent possible negative values in model predictions.

L236: What does “R” represent here? Is this an R2 of EPG vs actual worm burden for the sacrificed individuals? If so, perhaps use R2 instead of R. If this is meant to be the raw correlation, perhaps use “r” instead of “R” for consistency.

L237: Throughout the results, it would be helpful to see some measures of model fit in addition to the coefficients and p values. Can you provide the marginal and conditional R2?

Figure 1 is very helpful to visualize the laboratory experiment. Kudos to the authors for making such a figure in ggplot.

Figure 2 and 4: In the legend (diet), could you relabel this as “wild diet” and “laboratory diet” (as in Table 1) for the top and bottom rows? It wasn’t immediately clear to me (without reading the legend) what was the field experiment and what was the laboratory experiment. In the legend, perhaps note that points are jittered to reduce overlap.

Figure 3: Perhaps clarify in the legend that these data are from the field experiment.

Table S2 is very helpful for breaking down the model formulations.

Review form: Reviewer 2

Recommendation

Major revision is needed (please make suggestions in comments)

Scientific importance: Is the manuscript an original and important contribution to its field?
Excellent

General interest: Is the paper of sufficient general interest?
Good

Quality of the paper: Is the overall quality of the paper suitable?
Good

Is the length of the paper justified?
Yes

Should the paper be seen by a specialist statistical reviewer?
No

Do you have any concerns about statistical analyses in this paper? If so, please specify them explicitly in your report.
Yes

It is a condition of publication that authors make their supporting data, code and materials available - either as supplementary material or hosted in an external repository. Please rate, if applicable, the supporting data on the following criteria.

Is it accessible?
Yes

Is it clear?

Yes

Is it adequate?

Yes

Do you have any ethical concerns with this paper?

No

Comments to the Author

See attached file. (See Appendix A)

Decision letter (RSPB-2020-0859.R0)

18-May-2020

Dear Ms Sweeny:

I am writing to inform you that your manuscript RSPB-2020-0859 entitled "Supplemented nutrition decreases helminth burden and increases drug efficacy in a natural host-helminth system" has, in its current form, been rejected for publication in Proceedings B.

This action has been taken on the advice of referees, who have recommended that substantial revisions are necessary. With this in mind we would be happy to consider a resubmission, provided the comments of the referees are fully addressed. However please note that this is not a provisional acceptance.

Sincerely,

Dr Locke Rowe

Associate Editor

Comments to Author:

Thank you for submitting your manuscript “Supplemented nutrition decreases helminth burden and increases drug efficacy in a natural host-helminth system” To Proceedings B. I have now received two reviews and evaluated the manuscript myself. While we all find the topic important and the experiments well conducted and written up, several issues have been raised - mainly to do with contextualizing the topic and results - that should be addressed. For example, reviewer 2 raises several points around your expectations of the results and alternative scenarios. Reviewer 1 highlights how the impact of this study can be broadened by addressing the implications for wildlife health. Related to both points, more discussion of other food provisioning-pathogen/parasite experiments with different wildlife species would be beneficial.

Reviewer(s)' Comments to Author:

Referee: 1

Comments to the Author(s)

The authors here robustly combine a field and laboratory experiment to factorially test the effects of nutrient supplementation and parasite removal on host immunity and helminth outcomes. The authors demonstrate that supplemental nutrition rapidly improves host resistance to helminths, increases the efficacy of antihelminth treatment, and improves condition and immune defense. This is a very strong study, and I had relatively minor comments about the actual design and analysis. The authors are very transparent about their results, and it was nice to see all the model outputs provided in the supplement.

My only major comment is a more conceptual one. Currently, the manuscript is framed (L81-97) around helminth infections, malnutrition, and supplementation in humans; that laboratory mouse models can inform how nutrients affect infection outcomes; but that wild experiments are needed to better represent the co-evolutionary dynamics of hosts and parasites and natural conditions. Wild mice undoubtedly provide insight into how nutrition impacts parasite outcomes in natural populations, but it isn't entirely clear how that would then translate back to human interventions per se. In general, I think the authors could also add the justifying point (in both the Introduction and Discussion) that nutritional supplementation is also an intervention worth considering in wildlife themselves (e.g., for species of conservation concern). The results of the current study (rapid improvements to host resistance) also carry important implications for how supplemental food resources could affect infection in wild species that are reservoir hosts of zoonotic pathogens. Some discussion of these contexts would only broaden the relevance of the study for not only humans but also wildlife themselves.

Minor comments:

L126: Can you provide the months of the wood mouse breeding season?

L192: Replace with “Generalized Linear Mixed Models”

L194: Is the negative binomial appropriate for mean EPG? If this is a mean across captures, is it still count data (i.e., integers) or now continuous (i.e., with decimals)? If the resulting means fall into the latter, were numbers rounded to the nearest integer for the NB?

L213: Were mass, fat, IgA, and IgG log-transformed prior to the GLMMs? That would prevent possible negative values in model predictions.

L236: What does “R” represent here? Is this an R² of EPG vs actual worm burden for the sacrificed individuals? If so, perhaps use R² instead of R. If this is meant to be the raw correlation, perhaps use “r” instead of “R” for consistency.

L237: Throughout the results, it would be helpful to see some measures of model fit in addition to the coefficients and p values. Can you provide the marginal and conditional R2?

Figure 1 is very helpful to visualize the laboratory experiment. Kudos to the authors for making such a figure in ggplot.

Figure 2 and 4: In the legend (diet), could you relabel this as “wild diet” and “laboratory diet” (as in Table 1) for the top and bottom rows? It wasn’t immediately clear to me (without reading the legend) what was the field experiment and what was the laboratory experiment. In the legend, perhaps note that points are jittered to reduce overlap.

Figure 3: Perhaps clarify in the legend that these data are from the field experiment.

Table S2 is very helpful for breaking down the model formulations.

Referee: 2

Comments to the Author(s)

See attached file.

Author's Response to Decision Letter for (RSPB-2020-0859.R0)

See Appendix B.

RSPB-2020-2722.R0

Review form: Reviewer 2

Recommendation

Accept with minor revision (please list in comments)

Scientific importance: Is the manuscript an original and important contribution to its field?

Excellent

General interest: Is the paper of sufficient general interest?

Good

Quality of the paper: Is the overall quality of the paper suitable?

Excellent

Is the length of the paper justified?

Yes

Should the paper be seen by a specialist statistical reviewer?

No

Do you have any concerns about statistical analyses in this paper? If so, please specify them explicitly in your report.

No

It is a condition of publication that authors make their supporting data, code and materials available - either as supplementary material or hosted in an external repository. Please rate, if applicable, the supporting data on the following criteria.

Is it accessible?

Yes

Is it clear?

Yes

Is it adequate?

Yes

Do you have any ethical concerns with this paper?

No

Comments to the Author

I was reviewer #2 in the original version of this manuscript, and I want to thank Dr. Sweeny and colleagues for the thoughtful, detailed responses to my comments. I am further convinced that this is a novel, important study and I am glad to see it published in Proc B, where I think it will be an impactful study, as it clearly demonstrates the value of hard-won data on wild animals in furthering our understanding of infection and immunity in nature. I have only a few minor comments and questions that can be addressed fairly easily.

There is a result that I didn't pick up on in my first review of this paper, and almost missed this time, that I think might be worth emphasizing even more. This is the result in Fig. 2B (the post-treatment EPG). Fig. 2C shows the worm burdens for animals caught between 12-16 days post-drug-treatment, whereas Fig. 2B shows the EPG for animals caught *at any time* post-drug-treatment, including (presumably), times less than 12-16 days, but also including times greater than 16 days. (How many of the recaptures fall into either window is not currently clear.) If most of those recaptures are post-16 days, then doesn't this indicate that the treatment affected transmission, since you expected that any individual protection against reinfection would wear off after about 16 days, meaning that the continued low EPG suggests that reinfection is rare because transmission has been short-circuited? If so, I think that's something worth pointing out explicitly in the Results and Discussion.

My only other question of substance concerns the relationship between body condition, immunity, and infection. Fig. 4 shows the relationship between body condition and immunity, showing that animals that are in better condition (e.g., are heavier for the same length) have stronger immunity. But on lines 244-247, you say that larger mice had higher worm burdens, and you don't show the relationship between immunity and worm burden (or EPG). How should a reader make sense of these results? I don't think this requires a long explanation, but maybe some acknowledgment of this (seeming) discrepancy and a potential explanation in the Results section would be helpful.

Minor grammatical stuff:

"Pearsn's" instead of "Pearson's" in line 233

"anthelmintic" instead of "anhelminthic" in line 258

Missing a period at the end of line 368

Decision letter (RSPB-2020-2722.R0)

07-Dec-2020

Dear Ms Sweeny

I am pleased to inform you that your manuscript RSPB-2020-2722 entitled "Supplemented nutrition decreases helminth burden and increases drug efficacy in a natural host-helminth system" has been accepted for publication in Proceedings B.

The referee(s) have recommended publication, but also suggest some minor revisions to your manuscript. Therefore, I invite you to respond to the referee(s)' comments and revise your manuscript. Because the schedule for publication is very tight, it is a condition of publication that you submit the revised version of your manuscript within 7 days. If you do not think you will be able to meet this date please let us know.

Sincerely,
Dr Locke Rowe
mailto: proceedingsb@royalsociety.org

Associate Editor
Board Member
Comments to Author:

Thank you for addressing the reviewer comments so thoroughly. The manuscript is much improved. It has been seen again by one of the initial reviewers, who has noted two additional points related to EPG counts and the relationship among body condition, immunity and infection that should also be addressed.

Reviewer(s)' Comments to Author:
Referee: 2

Comments to the Author(s).

I was reviewer #2 in the original version of this manuscript, and I want to thank Dr. Sweeny and colleagues for the thoughtful, detailed responses to my comments. I am further convinced that this is a novel, important study and I am glad to see it published in Proc B, where I think it will be an impactful study, as it clearly demonstrates the value of hard-won data on wild animals in furthering our understanding of infection and immunity in nature. I have only a few minor comments and questions that can be addressed fairly easily.

There is a result that I didn't pick up on in my first review of this paper, and almost missed this time, that I think might be worth emphasizing even more. This is the result in Fig. 2B (the post-treatment EPG). Fig. 2C shows the worm burdens for animals caught between 12-16 days post-drug-treatment, whereas Fig. 2B shows the EPG for animals caught *at any time* post-drug-treatment, including (presumably), times less than 12-16 days, but also including times greater than 16 days. (How many of the recaptures fall into either window is not currently clear.) If most of those recaptures are post-16 days, then doesn't this indicate that the treatment affected transmission, since you expected that any individual protection against reinfection would wear off after about 16 days, meaning that the continued low EPG suggests that reinfection is rare because transmission has been short-circuited? If so, I think that's something worth pointing out explicitly in the Results and Discussion.

My only other question of substance concerns the relationship between body condition, immunity, and infection. Fig. 4 shows the relationship between body condition and immunity, showing that animals that are in better condition (e.g., are heavier for the same length) have stronger immunity. But on lines 244-247, you say that larger mice had higher worm burdens, and you don't show the relationship between immunity and worm burden (or EPG). How should a reader make sense of these results? I don't think this requires a long explanation, but maybe some acknowledge of this (seeming) discrepancy and a potential explanation in the Results section would be helpful.

Minor grammatical stuff:

"Pearsn's" instead of "Pearson's" in line 233

"anthelmintic" instead of "anhelminthic" in line 258

Missing a period at the end of line 368

Author's Response to Decision Letter for (RSPB-2020-2722.R0)

See Appendix C.

Decision letter (RSPB-2020-2722.R1)

15-Dec-2020

Dear Ms Sweeny

I am pleased to inform you that your manuscript entitled "Supplemented nutrition decreases helminth burden and increases drug efficacy in a natural host-helminth system" has been accepted for publication in Proceedings B.

Open Access

Paper charges

Sincerely,

Appendix A

Review of “Supplemented nutrition decreases helminth burden and increases drug efficacy in a natural host-helminth system” by Sweeny et al.

In this paper the authors supplemented the diet of both captive and wild populations of wood mice to investigate how diet influences (1) susceptibility to infection with the roundworm *Heligmosomoides polygyrus*; (2) efficacy of anthelmintic treatment; and (3) development of immunity to future infection. I found the use of both wild and laboratory populations of wood mice to be both unique and innovative. This combination really increases the strength of the conclusions drawn here, as either study on its own would have weaknesses that are well-addressed by the other component of the study. I have a few general comments about the broad sweep of the paper, and a number of specific comments and suggestions to improve the readability. My one criticism of the paper in its current form is that I found the description of, and justification for, the experimental design to be a bit lacking. However, I think this can be addressed fairly straightforwardly on revision. I think this article is well-suited to the readership of *Proc. B.* and will be a very useful case study for future experiments studying host-helminth interactions in natural populations.

General comments:

1. An obvious (albeit trivial) response to this article would be that these results are very intuitive. Obviously, that doesn't weaken the value of the study, but I'm wondering whether you had any expectation that supplemental nutrition might *not* have had these effects. Is there an argument that supplemental nutrition would do something other than what was observed here? Could dietary supplementation *increase* susceptibility, *decrease* drug efficacy, *weaken* antibody responses and *increase* parasite egg shedding? Presumably any such argument would be based on the idea that dietary supplementation could fuel parasite growth? Is that plausible, given the biology of *H. polygyrus*?

2. One of the things that is interesting about this study is that it is a whole-diet supplementation study, rather than being fixated on a particular macro- or micronutrient. This comes up in the Discussion, but I would be curious to see some more discussion of how the results of your experiment differ from those of the studies that manipulate specific nutrients. On lines 359-365, you discuss the fact that many of these studies use much more dramatic differences in nutrient availability, so are the results of your experiment surprising? That is, do you see differences that are larger (or smaller) than you might have expected? For example, if you saw a change in, e.g., antibody concentrations between control and supplemented that were of a similar magnitude to the differences observed between low and high protein diets, that would be interesting and would point to interactive effects of nutrients on condition/immunity/parasites.

3. This discussion might also help to address an implicit assumption that people might have that the reaction norms between, e.g., immunity and diet or parasite shedding and diet will be linear (e.g., if increasing total calories by 20% reduces egg shedding by 10% compared to the control, then decreasing total calories by 20% will increase egg shedding by 10%), and this is an opportunity to discuss whether this is the case in this system. Comparing your study with

energy restriction studies (e.g., Koski, Su, and Scott 1999 *Biochem. Biophys. Res. Commun.*, 264, 796-801), does supplementation enhance common response variables more than restriction hinders them (e.g., the reaction norm is concave up), or does restriction hinder more than supplementation helps (e.g., the reaction norm is concave down), or are the responses similar (e.g., the reaction norm is linear)?

4. I would like to see more discussion of the fact that food supplementation had an effect regardless of how long individuals spent on the supplemented plots (based on my interpretation of lines 200-203 and ESM Fig. S1). Is that because there were very few animals who moved between treatments (as far as you know), so your conclusions there are fairly weak, or do you think that even a small amount of dietary supplementation can have a large effect?

Specific comments:

Lines 62-64 and 67-68: While it is clear that control will require reducing exposure, this is not something your experiment can really address, and it is not clear to me that the processes that drive exposure in mice are necessarily similar to the processes that drive exposure in humans (whereas it is easier to imagine that the immune processes that determine susceptibility are similar).

Line 94-95: Can you make this sentence a bit more explicit about how your treatment differs from previous studies of dietary supplementation in wild mice? What does it mean to say that, “studies supplemented the food supply to mimic natural variation rather than increased food quality?” Does this mean they just added more of whatever was found in the environment, regardless of whether what was found was high quality or not? How are you assessing the quality of the dietary supplementation in those studies?

Line 95-97: Why are all of the studies referenced in this paragraph insufficient to help us develop an understanding of, and intuition for, how nutrition interacts with immunity? If I was the author of one of the more controlled studies using specific micro- or macronutrients (I’m not!), I might argue that this control gives me a better sense of how nutrition interacts with immunity. This might be a good place to talk about/foreshadow my comment #2 above: if results of whole diet supplementation cannot be predicted from the results of studies that supplement only specific micro- or macronutrients, that is a good rebuttal to my hypothetical argument.

Line 101-102: I’m assuming “commonly” in this sentence refers to high prevalence, and “chronically” refers to long infection duration? Is this really two pieces of data, or are you assuming chronic infections on the basis of high prevalence? That is, are there actually studies measuring infection duration in wild populations?

Line 104-107: I interpret “nutrition availability” as “availability in the environment,” so I don’t understand how simultaneous energetic demands should impact availability. Do you mean availability of nutrients within the host? Or is this a reference to the fact that, if foraging is

energetically costly and immunity is energetically costly, that mounting an immune response could reduce the energy available for foraging, which would then reduce the intake of nutrients?

Line 110: On the first readthrough, I didn't quite follow the structure of this list, as it was not clear whether these were three different experiments, or three different analyses done in on data from a single experiment? Afterwards, of course, I realized it was two experiments. It would be helpful to make the experimental design clear here, and also to explain why you did two experiments by explaining how these experiments (and the measurements undertaken in each) complement one another.

Line 124-130:

- Have you previously confirmed that food availability and infection (prevalence and/or intensity) were similar between grids prior to your experimental manipulations? This is critical for understanding whether your results are due to a treatment effect versus an intrinsic environmental difference. Also, how would you have described the food conditions on these plots prior to supplementation? That is, are the mice that live on these plots in fairly good condition (compared to whatever is normal for wood mice)? I am wondering especially since it seemed like any mouse that was ever found on a supplemented plot had better outcomes than the mice that were never found on a supplemented plot.
- How far apart are these grids from one another? How does the distance between plots compare to the distance wood mice travel within a day/week/month? Were you expecting that mice would travel between grids?

Lines 132-139 and 194: I am a bit uncertain about the experimental protocol here. On line 194, you measure the intensity of infection at first capture, which you say is "before treatment." Do you mean "before drug treatment" or do you mean "before diet or drug treatment"? Based on my reading of lines 132-135, it seems like diet treatments were started before the first trapping. I'm assuming this is just the ambiguity of the word "treatment" based on the rest of the manuscript.

Line 143: Would it be correct to say that any mouse caught 12-16 days after first capture was sacrificed, and any mouse caught outside that window was released (and thus only had non-destructive data collected from it)? I didn't read this sentence that way the first time through, and it wasn't until I got to lines 233-236 that I understood what you actually did.

Line 152-154: Why did you use two different diets for the laboratory experiment? Why not feed both diet treatments with Rat Mouse 1, and then supplement with Transbreed? This would have seemed to be more parallel to the field experiment protocol.

Line 192-203:

- I did not understand why age was only included in the worm burden model when I first read this section. It would be helpful to state on line 195 that infection burden and age

were only measured in the few animals that were captured between 12-16 days after initial capture.

- What was the distribution of time spent on the supplemented plots across all individuals (that is, were most of the animals described as “supplemented” found on the supplemented plots a lot, or were there a lot of animals found on supplemented plots ~50% of the time)? [From reading the ESM, I see that only 16% of animals were found on both plot types – I would move this information into the main text.]
 - o Now that I think about it, I’m not sure how you would even calculate the proportion of time that a mouse spends on a grid! Do you just mean the fraction of all trappings that occur on a supplemented grid, so an animal is classified into the “supplemented” treatment if 2 of the 3 times it was trapped it was on the supplemented grid? This question comes back below (line 239) in how you calculated how long each mouse spends on the supplemented grids.
- The description of the results of the model with three diet treatment levels is a bit ambiguous. You say that “effects of supplemented nutrition were not dependent on time spent on grid type.” Based on my reading of the ESM, it seems like what you mean here is that individuals who spent less than 100% on the supplemented grid were statistically indistinguishable in their measurements to individuals that spent 100% of time on the supplemented grid. Is this because there were only a few animals that spent time on both sites?
- If the grids were close enough to allow migration, were there trends in capture site observed over the course of the field trapping? That is, did you find that animals that were originally captured on control diet grids were more likely to be captured later on supplemented grids than vice versa?

Lines 225-227: What kind of regression did you do to calculate body condition (e.g., the regression between body weight and body length)?

Lines 229-231: This paragraph needs to be moved to beginning of section 2.4. I spent the whole section wondering whether you really did GLMMs or if you actually did GLMs, and how you accounted for the fact that you have repeated measures of the same individual. However, and I’ll admit that I’m out of my depth here, is including the individual mouse as a random effect the appropriate way to handle this data? For example, if individuals differ in their growth rate, then the difference in body mass from timepoint 1 to timepoint 2 will actually get larger, rather than remaining a fixed difference – I guess this comes down to a difference between a random intercepts model or a random slopes model? A bit more explanation of the statistical procedure would be helpful here.

Line 239: I don’t understand how you estimate the amount of time mice spent on supplemented grids. For example, if you caught a mouse a supplemented grid initially, caught it 20 days later on the supplemented grid, caught it 10 days after that on a control grid, and finally caught it 10 days after that on a supplemented grid, how many days would you say it spent on supplemented grids?

Fig. 2: How do you reconcile the fact that peak and total EPG is much lower in the secondary challenge for both control and supplemented animals, but that end point worm burden is higher for secondary challenge than primary challenge for the animals on the control diet? E.g., putting Fig 2E and Fig. 2F together suggests that control diet animals had more worms, but that those worms were reproducing much less. Were these worms particularly stunted or something?

Fig. 4: You might remind readers in the caption that body condition is the residual of a body weight/body length regression.

Grammatical stuff:

Line 68: comma instead of semicolon since you are using a conjunction

Line 78: no comma after “trade-offs”

Line 82: “has been explored” occurs twice in this sentence

Line 105: comma instead of semicolon since these are not independent clauses (or add “these are” before “conditions”)

Line 114: comma instead of semicolon since these are not independent clauses (or replace “suggesting” with “this suggests”)

Line 126: comma instead of semicolon since these are not independent clauses

Line 145: why is the word last in quotes here?

Line 151-152: the second clause of this sentence seems to be missing a word (maybe “were” before manipulated?)

Line 163: “samples” rather than “sample”

Line 164: “from day 0” is unnecessary in this sentence

Line 193: I think it would be clearer to say “Models fit to the data from the wild individuals” rather than “Wild models” since the models aren’t wild (a similar issue exists on line 205)

Line 196: Add “and” after the comma

Line 197: “fixed effects” rather than just “effects”

Line 222: ‘dpi’ has not been defined in the main text previously

Line 294: comma rather than semicolon since these are not independent clauses

Appendix B

To the editors of *Proceedings of the Royal Society B*,

We would like to thank you and both referees for their thorough and helpful reviews. We are confident that their suggestions have substantially improved the manuscript. As suggested, we have made several major revisions to the manuscript, each of these are highlighted below under the specific comments from the 2 referees (which are in bold). In particular, we made edits to more clearly communicate the experimental design and the framing of the results. We reference updated line numbers from the revised document for edits throughout and where appropriate provide updated wording for ease of review.

We hope that this revised and improved manuscript will be well-suited for *Proceedings for the Royal Society B*.

Kind regards,

Amy Sweeny

Referee 1:

My only major comment is a more conceptual one. Currently, the manuscript is framed (L81-97) around helminth infections, malnutrition, and supplementation in humans; that laboratory mouse models can inform how nutrients affect infection outcomes; but that wild experiments are needed to better represent the co-evolutionary dynamics of hosts and parasites and natural conditions. Wild mice undoubtedly provide insight into how nutrition impacts parasite outcomes in natural populations, but it isn't entirely clear how that would then translate back to human interventions per se. In general, I think the authors could also add the justifying point (in both the Introduction and Discussion) that nutritional supplementation is also an intervention worth considering in wildlife themselves (e.g., for species of conservation concern). The results of the current study (rapid improvements to host resistance) also carry important implications for how supplemental food resources could affect infection in wild species that are reservoir hosts of zoonotic pathogens. Some discussion of these contexts would only broaden the relevance of the study for not only humans but also wildlife themselves.

Thank you for this suggestion. We agree with the referee here, and much of our research in this area focuses on how nutrition supplementation has broad impacts on wildlife health and dynamics. We have therefore shifted the narrative of the introduction and discussion to reduce the emphasis on translation to human systems and highlight the implications that the impact of nutrition has on helminth infection for wildlife populations more broadly given their high rate of infection in wild populations and the rapidly changing landscape of resources available to wildlife. The introduction has been adjusted to capture these nuances and in particular, paragraphs 3-4 (lines 66-94) have been reframed to highlight the broader implications of resources. Specifically, we reference recent work by Daniel Becker and colleagues which highlights the complex potential of altered resources to influence wildlife health to emphasise the utility of experimental manipulation with novel diet for wildlife populations.

Minor comments:

L126: Can you provide the months of the wood mouse breeding season?

We have added details on the months describing the span of peak breeding season to this sentence (Page 5, line 123).

L192: Replace with “Generalized Linear Mixed Models”

This has been corrected in the text.

L194: Is the negative binomial appropriate for mean EPG? If this is a mean across captures, is it still count data (i.e., integers) or now continuous (i.e., with decimals)? If the resulting means fall into the latter, were numbers rounded to the nearest integer for the NB?

We apologise for the confusion here. Faecal samples for wood mice are most often <1g, and therefore counts of parasite eggs are divided by the weight of the sample to obtain an approximation of egg per gram faeces (EPG). These values can contain decimals, but represent parasite counts and are rounded to the nearest integer for models fit with negative-binomial error

distribution due to overdispersion. This has now been explicitly stated in the text lines 174-176 as follows:

“We measured H. polygyrus shedding as eggs per gram of faeces (EPG) using salt flotation [1] (details in ESM Section 1.3.1). Briefly, H. polygyrus eggs were counted and standardised by the weight of the sample to estimate EPG. EPG values were rounded to the nearest integer for subsequent analysis.”

L213: Were mass, fat, IgA, and IgG log-transformed prior to the GLMMs? That would prevent possible negative values in model predictions.

We have edited the text to improve clarity. Specifically, body mass followed a normal distribution and was not transformed; likewise, fat scores represent ordinal values and were not transformed for analysis. We used GLMMs with gaussian errors distributions and cumulative link mixed models for ordinal data for these metrics. This has now been more clearly stated (Page 7, lines: 213-215). For the antibody day, both IgA and IgG values represented standardised concentrations and were not transformed in the models. Although there were some negative model coefficients, model predictions for each of the above responses did not contain negative values nor negative intercepts (Table S8). However, when body condition index (BCI) was used as a predictor in the immunity models, BCI represented the regressions of a mass ~ length regression (Page 8, lines 227-228) and therefore there are negative values included in axes for Figure 4 where body condition was a significant predictor of both IgA and IgG1.

L236: What does “R” represent here? Is this an R² of EPG vs actual worm burden for the sacrificed individuals? If so, perhaps use R² instead of R. If this is meant to be the raw correlation, perhaps use “r” instead of “R” for consistency.

We apologise for this error, we should have used ‘r’ instead of ‘R’ as we were referring to Pearson’s correlation. This has now been corrected in the text (Page 8, line 233) and supplementary figure (Figure S3).

L237: Throughout the results, it would be helpful to see some measures of model fit in addition to the coefficients and p values. Can you provide the marginal and conditional R²?

Thank you for this suggestion. We agree this is helpful context and have now addressed this issue by adding both marginal and conditional R² values to the model results tables within the supplementary material for each respective model table. In some instances, we report only marginal R² as variance of random effects was negligible and conditional R² could not be estimated.

Figure 1 is very helpful to visualize the laboratory experiment. Kudos to the authors for making such a figure in ggplot.

Thanks, we are very glad this was of use!

Figure 2 and 4: In the legend (diet), could you relabel this as “wild diet” and “laboratory diet” (as in Table 1) for the top and bottom rows? It wasn’t immediately clear to me (without reading the legend) what was the field experiment and what was the laboratory experiment. In the legend, perhaps note that points are jittered to reduce overlap.

We thank the referee for this suggestion. The legends for both figures have been updated as requested and specification has been added to the legends to note that the points were jittered for visibility.

Figure 3: Perhaps clarify in the legend that these data are from the field experiment.

This has now been clarified in the legend.

Table S2 is very helpful for breaking down the model formulations.

Much appreciated!

Referee 2

General comments:

1. An obvious (albeit trivial) response to this article would be that these results are very intuitive. Obviously, that doesn't weaken the value of the study, but I'm wondering whether you had any expectation that supplemental nutrition might not have had these effects. Is there an argument that supplemental nutrition would do something other than what was observed here? Could dietary supplementation increase susceptibility, decrease drug efficacy, weaken antibody responses and increase parasite egg shedding? Presumably any such argument would be based on the idea that dietary supplementation could fuel parasite growth? Is that plausible, given the biology of *H. polygyrus*?

We think the referee brings up an interesting point here – that the results are intuitive and predictable. We would argue that although results may align with intuition post-hoc, this was not a forgone conclusion. There is a growing body of literature which highlights the diverse range of outcomes possible in wildlife following supplemented resources. A framework by Becker *et al.* 2015 (*Ecology Letters*) offers multiple mechanisms by which nutrition may interact with host infection given the multi-faceted effects on host condition, behaviour, and demography. Therefore although it is possible that nutrition would reduce infection burdens physiologically via improved condition and immunity, this requires explicit testing in the wild as numerous studies have documented food supplementation in the wild as detrimental to animal health and causative of increased parasite infection due to changes in behaviour and aggregation around food sources via meta-analysis, empirical data, and modelling (eg Becker *et al.* 2015, Becker *et al.* 2014, Moyers *et al.* 2018; now cited in discussion). Several additional bodies of work highlight nuance in the host-resource relationship with regard to immunity and infection. For example, Standin *et al.* (2018) reviewed effects of resource provisioning on host immunity and found varied results across captive versus field populations and according to nutrient type and van Leeuwen *et al.* (2019) propose a theoretical framework showing that resource flows within host can drive divergent outcomes in helminth infections.

Our motivation for this study was to test the impact of supplemental nutrition using the same natural host-parasite system in both the wild and a controlled laboratory setting – so that we could begin to disentangle the many ways in which resources may impact host parasite interactions. Previous evidence in model systems suggests that improved nutrition increases

immunity to *H. polygyrus* specifically (referenced throughout manuscript), but whether this would translate to the wild setting was as yet unclear. We believe that this paired wild-laboratory approach is a key novelty of the study in this regard, and we have made some minor edits to the introduction and discussion to make our predictions and the implications for our results clearer. Specifically, we have now added changes in the introduction (Page 4, lines 86-94) and edited sections of the discussion to more adequately acknowledge this nuance. In particular we have broadened our referencing in the discussion to highlight where there have been (and could be) unintended outcomes of the nutrition- immunity relationship and on Page 13, lines 335-348 we discuss the results from this experiment in the context of the theoretical hypotheses that may underlie resource supplementation effects on helminth infections.

2. One of the things that is interesting about this study is that is a whole-diet supplementation study, rather than being fixated on a particular macro- or micronutrient. This comes up in the Discussion, but I would be curious to see some more discussion of how the results of your experiment differ from those of the studies that manipulate specific nutrients. On lines 359- 365, you discuss the fact that many of these studies use much more dramatic differences in nutrient availability, so are the results of your experiment surprising? That is, do you see differences that are larger (or smaller) than you might have expected? For example, if you saw a change in, e.g., antibody concentrations between control and supplemented that were of a similar magnitude to the differences observed between low and high protein diets, that would be interesting and would point to interactive effects of nutrients on condition/immunity/parasites.

As noted by the referee, our aim here was to investigate the impact of whole-diet supplementation as opposed to taking a more targeted-nutrient approach. We agree that the discussion would benefit from further acknowledgement of the impact of diet on antibody concentrations and how this compares to other studies which specifically manipulated one aspect of diet (e.g. protein). While it is difficult to compare antibody responses across systems, we have now incorporated more discussion and comparison of our results (Page 14, lines 353-367 and lines 369-378). We use caution in direct comparisons, citing work in nutritional geometry highlighting the multi-dimensional nature of macro & micro nutrient effects, but have provided additional context on how we interpret results of the whole-diet supplementation.

3. This discussion might also help to address an implicit assumption that people might have that the reaction norms between, e.g., immunity and diet or parasite shedding and diet will linear (e.g., if increasing total calories by 20% reduces egg shedding by 10% compared to the control, then decreasing total calories by 20% will increase egg shedding by 10%), and this is an opportunity to discuss whether this is the case in this system. Comparing your study with energy restriction studies (e.g., Koski, Su, and Scott 1999 *Biochem. Biophys. Res. Commun.*, 264, 796-801), does supplementation enhance common response variables more than restriction hinders them (e.g., the reaction norm is concave up), or does restriction hinder more than supplementation helps (e.g., the reaction norm is concave down), or are the responses similar (e.g., the reaction norm is linear)?

We agree that using reaction norms context would be a useful perspective. However, we are cautious of over-interpreting the comparison between our experiment and more restrictive

studies, and would not extrapolate the shape of the reaction norm given we could not measure consumption of supplementary food in the wild and all mice were fed libitum in the laboratory. Given the large effects in the laboratory despite modest diet group differences and rapid response in the wild, we interpret this more along the lines of supplementation enhancing more than restriction hinders, and have added additional discussion to this effect cautiously indicating this high magnitude of effect relative to the nutritional content (Page 14, lines 361-367). We believe that further work in this area using controlled diet experiments in the lab would be a good way to address this further in future.

4. I would like to see more discussion of the fact that food supplementation had an effect regardless of how long individuals spent on the supplemented plots (based on my interpretation of lines 200-203 and ESM Fig. S1). Is that because there were very few animals who moved between treatments (as far as you know), so your conclusions there are fairly weak, or do you think that even a small amount of dietary supplementation can have a large effect?

We do interpret our results (Page 8, lines 242-244 and Figs S1&S4 in revised MS) to suggest that a small amount of dietary supplementation can have a large effect on wood mice. However, we would like to use caution here as we will not be able to determine the exact amount of time or resources that a mouse would have had access too. We have clarified our discussion of this result (Page 14, lines 364-367). Additionally, we provide contextual information and a figure below in our responses below (Pages 9;11-12) to related queries regarding the supplement categories and grid movement.

Specific comments:

Lines 62-64 and 67-68: While it is clear that control will require reducing exposure, this is not something your experiment can really address, and it is not clear to me that the processes that drive exposure in mice are necessarily similar to the processes that drive exposure in humans (whereas it is easier to imagine that the immune processes that determine susceptibility are similar).

We apologise for the lack of clarity here, we have removed lines 67-68 as referred to here and rewritten this paragraph of the introduction to improve clarity. We agree that all populations are very different, but meant only to emphasise that any reduction of infectious material in the environment will reduce exposure to nematodes as exposure will be driven by contact with larvae.

Line 94-95: Can you make this sentence a bit more explicit about how your treatment differs from previous studies of dietary supplementation in wild mice? What does it mean to say that, “studies supplemented the food supply to mimic natural variation rather than increased food quality?” Does this mean they just added more of whatever was found in the environment, regardless of whether what was found was high quality or not? How are you assessing the quality of the dietary supplementation in those studies?

We have edited this text to improve clarity. We now highlight that many wild mouse supplementation studies (our work included) have supplemented with resources that are naturally available (e.g. acorns/seeds) and contrast this to our present study which introduced a novel diet with a range of extra micro-and macronutrients. Our use of ‘quality’ here was to

represent this diverse set of nutrients available. We have therefore revised the text to read as follows (Page 4, lines 90-92) :

“Although supplementation experiments have also been investigated in wild mouse models [7,34,35], these studies augmented resources of the same type as was available in the environment (eg acorns or seeds) rather than introducing supplemental food with additional micro- and macronutrients.”

Line 95-97: Why are all of the studies referenced in this paragraph insufficient to help us develop an understanding of, and intuition for, how nutrition interacts with immunity? If I was the author of one of the more controlled studies using specific micro- or macronutrients (I’m not!), I might argue that this control gives me a better sense of how nutrition interacts with immunity. This might be a good place to talk about/foreshadow my comment #2 above: if results of whole diet supplementation cannot be predicted from the results of studies that supplement only specific micro- or macronutrients, that is a good rebuttal to my hypothetical argument.

We agree with the referee that studies of specific micro- and macronutrients offer more control for understanding mechanistic relationships; however, motivation for this study does stem from the inability to translate predictions to whole-diet supplementation and/or wild populations as the referee indicates in the last sentence. Our emphasis in these lines was ‘in natural populations’ as phenotypic responses can vary dramatically between captive and natural conditions. We believe that the rewrite of the paragraph in which these lines appear now more accurately conveys this context & motivation (Pages 3-4, lines 79-94).

Line 101-102: I’m assuming “commonly” in this sentence refers to high prevalence, and “chronically” refers to long infection duration? Is this really two pieces of data, or are you assuming chronic infections on the basis of high prevalence? That is, are there actually studies measuring infection duration in wild populations?

We use these terms as they are used in the disease ecology literature, where ‘chronic’ refers to a long duration of infection within an individual and ‘common’ refers to a high prevalence in the population. We have investigated this host-parasite system in nature for >10 years and have evidence that infections are both chronic and common.

Line 104-107: I interpret “nutrition availability” as “availability in the environment,” so I don’t understand how simultaneous energetic demands should impact availability. Do you mean availability of nutrients within the host? Or is this a reference to the fact that, if foraging is energetically costly and immunity is energetically costly, that mounting an immune response could reduce the energy available for foraging, which would then reduce the intake of nutrients?

Here, we aim to convey the latter of the two interpretations, where simultaneous energetic demands can influence the allocation of nutrients. We have edited this sentence for clarity (Page 4, lines 101-104):

‘Further, wood mice, like most wild animals, have substantial and simultaneous energetic demands for reproduction, foraging, and survival [46,47], conditions which laboratory settings cannot replicate, but which likely impact infection exposure, immunity, and resource allocation’

Line 110: On the first readthrough, I didn't quite follow the structure of this list, as it was not clear whether these were three different experiments, or three different analyses done in on data from a single experiment? Afterwards, of course, I realized it was two experiments. It would be helpful to make the experimental design clear here, and also to explain why you did two experiments by explaining how these experiments (and the measurements undertaken in each) complement one another.

We have amended that list of experiment variables for clarity to the following two sentences (Page 4, lines 106-110):

“We used paired experiments in wild and laboratory populations of the same species to test the effects of supplemented nutrition and anthelmintic treatment on (i) H. polygyrus burden and egg shedding and (ii) body condition and immune responses. We use data from the laboratory population to better infer mechanisms in a controlled setting, and data from the wild for the translation to an ecologically realistic setting.”

Line 124-130:

- **Have you previously confirmed that food availability and infection (prevalence and/or intensity) were similar between grids prior to your experimental manipulations? This is critical for understanding whether your results are due to a treatment effect versus an intrinsic environmental difference. Also, how would you have described the food conditions on these plots prior to supplementation? That is, are the mice that live on these plots in fairly good condition (compared to whatever is normal for wood mice)? I am wondering especially since it seemed like any mouse that was ever found on a supplemented plot had better outcomes than the mice that were never found on a supplemented plot.**

We choose our grids to act replicates within a woodland – so we choose habitats that are as similar as possible. However, as with most ecological replicates there are slight vegetation differences between grids; however, all grids had comparable and significant tree cover. A large portion of wood mouse diet is comprised of tree mast and therefore yield is expected to be similar between grids. Due to the need for each mouse to have been supplemented for a minimum amount of time prior to our trapping and measurements we did not trap and assess infection prior to supplementation. However, we are confident that we are not detecting grid (or environmental) condition effects versus the effects of supplementation for several reasons. First, in each experiment year we use two grid replicates for each supplement group. Second, between years we swap which grids are designated as control and supplemented, such that if there are intrinsic environmental differences between grids they are not concentrated in one supplement group. In each model we include a grid:year random effect and find negligible variance associated with this term in all cases. Finally, with regard to baseline food conditions, we confirmed context of the natural food availability for each year using Nature's Calendar records describing the mast (tree fruit) scores for surrounding woodland. These should be interpreted with caution as they represent a large surrounding area, however the mast during this study was below average compared to later years within this system (2017, not included in this study). We have evidence from other studies on this field site that there is a positive correlation between mast scores and mouse baseline body condition. Below is the fruit score for Falkirk for temporal replicates represented in this experiment and the subsequent year for additional context.

- **How far apart are these grids from one another? How does the distance between plots compare to the distance wood mice travel within a day/week/month? Were you expecting that mice would travel between grids?**

Grids are spaced a minimum of 50m apart to reduce movement between grids as much as possible given the size of the woodland available; due to word count limitations this information is in the Supplementary Material. This distance is greater than the average home range of a wood mouse. Typically, we expect wood mice have a home range on the order of ~15m radius, where females have a slightly smaller home range compared to males. We observed through many years of trapping both in England and Scotland that wood mice tend to fall within these home range expectations and further have very high site and even trap fidelity. When released after trapping mice are returned to their exact capture location to minimise excess movement. Relative to our total numbers the 16 mice which have been trapped on >1 grid have only been found on a different grid for one of multiple captures, but do spend the majority of their time (inferred from capture locations) on a single grid. Among these animals who appear to travel between grids at all, 70% were male. Young males have the highest likelihood of moving outside their birth location so this is consistent with what we know of wood mice, and does not suggest movement between grids due to experimental supplementation.

Lines 132-139 and 194: I am a bit uncertain about the experimental protocol here. On line 194, you measure the intensity of infection at first capture, which you say is “before treatment.” Do you mean “before drug treatment” or do you mean “before diet or drug treatment”? Based on my reading of lines 132-135, it seems like diet treatments were started before the first trapping. I’m assuming this is just the ambiguity of the word “treatment” based on the rest of the manuscript.

We apologise for the confusion; ‘treatment’ here should represent drug treatment as only diet treatment was ongoing prior to first capture. This has been clarified in the manuscript to specify ‘drug treatment’.

Line 143: Would it be correct to say that any mouse caught 12-16 days after first capture was sacrificed, and any mouse caught outside that window was released (and thus only had nondestructive data collected from it)? I didn't read this sentence that way the first time through, and it wasn't until I got to lines 233-236 that I understood what you actually did.

This is the correct interpretation and we have clarified this as follows (Page 5, lines 140-142):

'Mice captured 12-16 days after first capture (the period of efficacy for this drug combination that we have previously observed in wild wood mice [52,55]) were sacrificed for additional destructive sampling. Mice caught beyond this date range, or those pregnant or lactating, were not sacrificed.'

Line 152-154: Why did you use two different diets for the laboratory experiment? Why not feed both diet treatments with Rat Mouse 1, and then supplement with Transbreed? This would have seemed to be more parallel to the field experiment protocol.

Both diets used in the laboratory are produced by the same company (SDS Special diet services). Rat Mouse 1 is a standard or maintenance chow while Transbreed chow is specifically formulated with higher quality nutrients for improved breeding success. Both diets, however, include the same core nutrient composition, so we do interpret Transbreed as having supplemental qualities to the same baseline makeup as RM1.

Line 192-203:

- **I did not understand why age was only included in the worm burden model when I first read this section. It would be helpful to state on line 195 that infection burden and age were only measured in the few animals that were captured between 12-16 days after initial capture.**

This line has been modified to read as follows for clarity (Page 7, lines 198-200):

'Age was only included as an explanatory variable in the worm burden model for sacrificed animals, where eye lens weight allowed estimation of age [57].'

- **What was the distribution of time spent on the supplemented plots across all individuals (that is, were most of the animals described as "supplemented" found on the supplemented plots a lot, or were there a lot of animals found on supplemented plots ~50% of the time)? [From reading the ESM, I see that only 16% of animals were found on both plot types – I would move this information into the main text.]**
 - o **Now that I think about it, I'm not sure how you would even calculate the proportion of time that a mouse spends on a grid! Do you just mean the fraction of all trappings that occur on a supplemented grid, so an animal is classified into the "supplemented" treatment if 2 of the 3 times it was trapped it was on the supplemented grid? This question comes back below (line 239) in how you calculated how long each mouse spends on the supplemented grids.**

Generally, mice described as ‘supplemented’ or ‘control’ were captured 100% percent of the time on the corresponding grid type. As described in the ESM for those 16 animals which were captured on both grid types, we assigned supplement category based on the grid type on which that mouse was found the majority of the time. We have updated the main text to specify the number of mice found on both grid types (Page 7, lines 202-203). We include a plot below in this response (Page 12) to lend greater clarity to those animals which were found on both grid types in response to all three bullet points raised within this comment. We acknowledge that time ‘time’ is an approximation here as we infer this given the fraction of captures on each grid type and have added this caveat to the text (line 203). Given we trapped 3x a week however, we feel that capture location is a very good approximation of where individuals are likely to be spending their time.

- **The description of the results of the model with three diet treatment levels is a bit ambiguous. You say that “effects of supplemented nutrition were not dependent on time spent on grid type.” Based on my reading of the ESM, it seems like what you mean here is that individuals who spent less than 100% on the supplemented grid were statistically indistinguishable in their measurements to individuals that spent 100% of time on the supplemented grid. Is this because there were only a few animals that spent time on both sites?**

As detailed above, our assignment of the small number of animals that spent time on both grid types to one of two supplement types does not influence results, and of biological interest raised in the discussion that effect of supplementation is not contingent on 100% of captures occurring on supplemented grids. We believe that this lack of statistical difference between ‘Mix’ and ‘Supplemented’ categories is due both to the small number of animals in the Mix category and an effect of even a small exposure to nutrition supplementation given the relatively low availability of resources in the wild. As highlighted below, any ‘grid movers’ were largely captured on one grid, with ~1-2 captures on the other.

- **If the grids were close enough to allow migration, were there trends in capture site observed over the course of the field trapping? That is, did you find that animals that were originally captured on control diet grids were more likely to be captured later on supplemented grids than vice versa?**

As raised above, grids were set-up outside the range expected for the vast majority of wood mouse movement and we are confident that the few mice that did move grids were controlled for in our analysis and are not impact our results or conclusions. We therefore do not believe that the majority of individuals found infrequently on other grid types represent a migration due to movement to supplemented food availability. Of the 16 animals shown below, only 5 represent individuals that were found primarily on control grids and sometimes on supplemented, and of these there was not a consistent or sustained direction of movement from control to supplemented grids. Rather, some were found first on supplemented and then for the remainder of captures on control grids which may indicated expected movement of young males from birth place to another resident location. Similarly, the one individual which does show sustained change of grid type from control to supplemented grid type (ID 15570566) was a young male at first capture as well.

Lines 225-227: What kind of regression did you do to calculate body condition (e.g., the regression between body weight and body length)?

Both body weight and length are normally distributed variables and body condition was calculated using a linear regression. This has been clarified on lines 227-228, Page 8.

Lines 229-231: This paragraph needs to be moved to beginning of section 2.4. I spent the whole section wondering whether you really did GLMMs or if you actually did GLMs, and how you accounted for the fact that you have repeated measures of the same individual. However, and I'll admit that I'm out of my depth here, is including the individual mouse as a random effect the appropriate way to handle this data? For example, if individuals differ in their growth rate, then the difference in body mass from timepoint 1 to timepoint 2 will actually get larger, rather than remaining a fixed difference – I guess this comes down to a difference between a random intercepts model or a random slopes model? A bit more explanation of the statistical procedure would be helpful here.

This paragraph has been moved to the beginning of section 2.4 for clarity, as suggested. We include ID as a random effect to avoid pseudoreplication (random-intercept model). The vast majority of captures and individuals in this dataset represent adults as we only tag beyond a certain weight threshold, so we do not expect that the timepoints and individuals contained represent growth phases with regard to body size. That said, we did initially confirm via analyses of adults only that the inclusion of subadults in the models as they are specified does not fundamentally alter the results due to possible growth rate confounding, so we are confident including random ID in this way deals with the data appropriately.

Line 239: I don't understand how you estimate the amount of time mice spent on supplemented grids. For example, if you caught a mouse a supplemented grid initially, caught it 20 days later on the supplemented grid, caught it 10 days after that on a control grid, and finally caught it 10 days after that on a supplemented grid, how many days would you say it spent on supplemented grids?

As detailed above, we infer time spent on grids roughly based on the proportion of captures. We have clarified the text (Page 7, lines 200-201) to read '*Diet were classified as 'supplemented' if > 50% of captures were on supplemented grids and as 'control' otherwise.*'

Fig. 2: How do you reconcile the fact that peak and total EPG is much lower in the secondary challenge for both control and supplemented animals, but that end point worm burden is higher for secondary challenge than primary challenge for the animals on the control diet? E.g., putting Fig 2E and Fig. 2F together suggests that control diet animals had more worms, but that those worms were reproducing much less. Were these worms particularly stunted or something?

We show the primary and secondary challenge side-by-side in panel F for comparison purposes between diets, however, we caution any direct comparison of worm burdens between the two, as they are not the same animals (Figure 1). Due to the destructive nature of worm burden quantification in the gut and as specified in the legend, the 'primary' challenge in panel F represents a group of mice which only ever received a primary challenge before sacrifice, while the secondary challenge received both. We chose to carry out worm burden counts in the primary only group as such to enable insight into differences between the two diet groups in this way. However, we do generally interpret the fact that secondary challenged individuals have worms present but little-to-no shedding as indicative that the host response to a secondary challenge after prior exposure in some way reduces the fecundity of the worms. Whether this is due to immune-mediated stunting of the worms, impediment to copulation, or sequestration of eggs has not been examined in this experiment, though in light of the extensive literature on nematode development, the first hypothesis seems more plausible.

Fig. 4: You might remind readers in the caption that body condition is the residual of a body weight/body length regression.

Clarification has been added to the figure legend.

Grammatical stuff:

Line 68: comma instead of semicolon since you are using a conjunction

Line 78: no comma after "trade-offs"

Line 82: "has been explored" occurs twice in this sentence

Line 105: comma instead of semicolon since these are not independent clauses (or add "these are" before "conditions")

Line 114: comma instead of semicolon since these are not independent clauses (or replace "suggesting" with "this suggests")

Line 126: comma instead of semicolon since these are not independent clauses

Line 145: why is the word last in quotes here?

Line 151-152: the second clause of this sentence seems to be missing a word (maybe "were" before manipulated?)

Line 163: “samples” rather than “sample”

Line 164: “from day 0” is unnecessary in this sentence

Line 193: I think it would be clearer to say “Models fit to the data from the wild individuals” rather than “Wild models” since the models aren’t wild (a similar issue exists on line 205) < come back to this >

Line 196: Add “and” after the comma

Line 197: “fixed effects” rather than just “effects”

Line 222: ‘dpi’ has not been defined in the main text previously

Line 294: comma rather than semicolon since these are not independent clauses

All minor comments listed above were corrected in the manuscript.

Appendix C

To the editors of *Proceedings of the Royal Society B*,

We would like to thank you and the referee for their helpful review of this revised manuscript. We are very pleased that the revisions improved the manuscript and have responded to remaining referee comments to improve clarity of the points raised. Below, we have responded to each referee comment (bolded) with our response and corresponding changes in the manuscript.

We believe this manuscript now meets all revision requests and will be well-suited for *Proceedings of the Royal Society B*.

Kind regards,

Amy Sweeny

Reviewer(s)' Comments to Author:

Referee: 2

Comments to the Author(s).

I was reviewer #2 in the original version of this manuscript, and I want to thank Dr. Sweeny and colleagues for the thoughtful, detailed responses to my comments. I am further convinced that this is a novel, important study and I am glad to see it published in Proc B, where I think it will be an impactful study, as it clearly demonstrates the value of hard-won data on wild animals in furthering our understanding of infection and immunity in nature. I have only a few minor comments and questions that can be addressed fairly easily.

We are very glad that our responses were helpful and appreciate the referee saying this! We have responded below to the remaining points raised.

There is a result that I didn't pick up on in my first review of this paper, and almost missed this time, that I think might be worth emphasizing even more. This is the result in Fig. 2B (the post-treatment EPG). Fig. 2C shows the worm burdens for animals caught between 12-16 days post-drug-treatment, whereas Fig. 2B shows the EPG for animals caught *at any time* post-drug-treatment, including (presumably), times less than 12-16 days, but also including times greater than 16 days. (How many of the recaptures fall into either window is not currently clear.) If most of those recaptures are post-16 days, then doesn't this indicate that the treatment affected transmission, since you expected that any individual protection against reinfection would wear off after about 16 days, meaning that the continued low EPG suggests that reinfection is rare because transmission has been short-circuited? If so, I think that's something worth pointing out explicitly in the Results and Discussion.

We thank the reviewer for raising this point and agree that reduced transmission is of great interest to us and we have carried out longer-term supplementation experiments to the effect of investigating these reinfection dynamics. However, Figure 2B does represent only time points at all captures *up until* 16 days post treatment. Given most animals recaptured 16 days or more were sacrificed at days 12-16 if possible (save for pregnant females), group sizes were very small beyond day 16 post-treatment. We therefore restricted data to this "endpoint" of the experiment. We have now edited the text in the methods to avoid any confusion and make this more explicit (lines 188-190):

"Because few mice were captured beyond the 12-16 day range for endpoint, and those that were were skewed toward reproductive females who were not sacrificed, EPG data for post-treatment captures was restricted to timepoints up to 16 days post-treatment."

My only other question of substance concerns the relationship between body condition, immunity, and infection. Fig. 4 shows the relationship between body condition and immunity, showing that animals that are in better condition (e.g., are heavier for the same length) have stronger immunity. But on lines 244-247, you say that larger mice had higher worm burdens, and you don't show the relationship between immunity and worm burden (or EPG). How should a reader make sense of these results? I don't think this requires a long explanation, but maybe some acknowledge of this (seeming) discrepancy and a potential explanation in the Results section would be helpful.

We agree this is a salient point. Where we indicate that larger animals had higher worm burdens, we believe that weight as a fixed effect is correlated with mouse age and represents a trend we find generally with this system where older mice have higher burdens due to higher exposure. However, in condition models the predictors for weight as a response as a measure of body condition include factors to control for age and overall size, particularly body length and reproductive status. In immune models, the body condition variable is scaled to host length and therefore represents a represents condition scaled to host size. We do include *H. polygyrus* infection as a predictor for immune models however we do not find that they are significantly correlated. We believe that this may be explained due to a high degree of variation in exposure history in the wild, and this was one reason we were motivated to investigate the relationship in the laboratory. We do note that in worm burden models we do note that animals with higher mass have higher burdens when age is also accounted for. Because this is number of worms we believe that this does not pose a discrepancy with higher condition increasing immune response given the immune response may be acting on the fecundity (EPG) and not the worm burden. However, we do acknowledge this can be confusing and we have made edits in several places to increase clarity and avoid misleading interpretation of results.

In the methods we have amended the description of the model effects dealing with age to read:

“Age was only included as an explanatory variable in the worm burden model for sacrificed animals, where eye lens weight allowed estimation of age [45]; in EPG models body mass was included as a less-resolved approximation of age” (lines 192-194)

“Reproductive status and body length were included as covariates in wild models to account for variation in body size. We used the same fixed effects for the laboratory models as in the wild with the exception of reproductive status which was not applicable and body length as absolute age was available” (lines 211-213)

In the discussion we have added the following caveat when we discuss condition and immunity results:

“Therefore, we suggest that the improved body condition of supplemented individuals may result in an indirect effect of supplementation on antibody levels and increased helminth resistance. Although mice with higher mass had higher worm burdens at final capture in the wild, results from the laboratory population suggest that nutrition effects on host response greatly reduces EPG even when worms are present.” (lines 354-358)

Minor grammatical stuff:

"Pearsn's" instead of "Pearson's" in line 233

This change has been made in the text.

"anthelmintic" instead of "anhelminthic" in line 258

This is the correct spelling and has been checked in the text.

Missing a period at the end of line 368

This change has been made in the text.